# Strongly Correlated Quantum Spin Liquids versus Heavy Fermion Metals: A Review

**DOI:** 10.3390/ma15113901

**Published:** 2022-05-30

**Authors:** Vasily R. Shaginyan, Alfred Z. Msezane, George S. Japaridze, Stanislav A. Artamonov, Yulya S. Leevik

**Affiliations:** 1Petersburg Nuclear Physics Institute, NRC Kurchatov Institute, 188300 Gatchina, Russia; start@pnpi.spb.ru; 2Clark Atlanta University, Atlanta, GA 30314, USA; amsezane@cau.edu (A.Z.M.); george.japaridze@gmail.com (G.S.J.); 3National Research University Higher School of Economics, 194100 St. Petersburg, Russia; ysl1968@mail.ru

**Keywords:** fermion condensation, topological quantum phase transition, flat band, quantum spin liquid, heavy fermion compound, frustrated compound

## Abstract

This review considers the topological fermion condensation quantum phase transition (FCQPT) that explains the complex behavior of strongly correlated Fermi systems, such as frustrated insulators with quantum spin liquid and heavy fermion metals. The review contrasts theoretical consideration with recent experimental data collected on both heavy fermion metals (HF) and frustrated insulators. Such a method allows to understand experimental data. We also consider experimental data collected on quantum spin liquid in ${\mathrm{Lu}}_{3}{\mathrm{Cu}}_{2}{\mathrm{Sb}}_{3}O_{14}$ and quasi-one dimensional (1D) quantum spin liquid in both ${\mathrm{YbAlO}}_{3}$ and $\mathrm{Cu}\left(C_{4}H_{4}N_{2}\right){\left({\mathrm{NO}}_{3}\right)}_{2}$ with the aim to establish a sound theoretical explanation for the observed scaling laws, Landau Fermi liquid (LFL) and non-Fermi-liquid (NFL) behavior exhibited by these frustrated insulators. The recent experimental data on the heavy-fermion metal α$-{\mathrm{YbAl}}_{1-x}{\mathrm{Fe}}_{x}B_{4}$, with x=0.014, and on its sister compounds β$-{\mathrm{YbAlB}}_{4}$ and ${\mathrm{YbCo}}_{2}{\mathrm{Ge}}_{4}$, carried out under the application of magnetic field as a control parameter are analyzed. We show that the thermodynamic and transport properties as well as the empirical scaling laws follow from the fermion condensation theory. We explain how both the similarity and the difference in the thermodynamic and transport properties of α$-{\mathrm{YbAl}}_{1-x}{\mathrm{Fe}}_{x}B_{4}$ and in its sister compounds β$-{\mathrm{YbAlB}}_{4}$ and ${\mathrm{YbCo}}_{2}{\mathrm{Ge}}_{4}$ emerge, as well as establish connection of these (HF) metals with insulators ${\mathrm{Lu}}_{3}{\mathrm{Cu}}_{2}{\mathrm{Sb}}_{3}O_{14}$, $\mathrm{Cu}\left(C_{4}H_{4}N_{2}\right){\left({\mathrm{NO}}_{3}\right)}_{2}$ and ${\mathrm{YbAlO}}_{3}$. We demonstrate that the universal LFL and NFL behavior emerge because the HF compounds and the frustrated insulators are located near the topological FCQPT or are driven by the application of magnetic fields.

## 1. Introduction

Strongly correlated Fermi systems are represented by numerous heavy fermion (HF) compounds characterized by their diverse microscopic properties. To study these strongly correlated Fermi systems, we employ a topological symmetry representing a powerful method for gaining knowledge about physical systems spanning from solids to galaxies and their clusters in the Universe [1,2]. Understanding of such symmetry and conditions for its violation allows one to obtain a general information about physical systems. The low-temperature universal properties of strongly correlated systems, including HF metals and frustrated insulators, can be unveiled within the fermion condensation (FC) theory [1,2,3,4,5,6,7]. This universality suggests that strongly correlated systems, or HF compounds, represent a new state of matter. It means that this new state is independent of the atomic composition of HF compounds, exhibits universal properties, and is defined by the formation of flat or approximately flat bands [1,2,3,4,5,6,7]. These bands, predicted many years ago [3,5,6] and discovered recently in graphene, see, e.g., [8,9,10], originate from a specific quantum phase transition known as the topological fermion-condensation quantum phase transition (FCQPT) that rearranges the Fermi surface into the Fermi volume, generating a flat band. Thus, for very different substances and under very different external conditions the universal topological FCQPT occurs at microscopic level, determining the macroscopic properties and universal behavior of HF compounds. These compounds proliferated and they include HF metals, quantum spin liquids, quasicrystals and two dimensional systems like ${}^{3}$He [1]. These HF compounds represent the new state of matter, since their behavior near the topological FCQPT acquires important similarities that make them universal [1,6,7,11,12,13,14].

In our brief review we consider recent experimental data collected on the frustrated insulator like ${\mathrm{Lu}}_{3}{\mathrm{Cu}}_{2}{\mathrm{Sb}}_{3}O_{14}$ and quasi-1D quantum spin liquid (1DQSL) in both ${\mathrm{YbAlO}}_{3}$ and $\mathrm{Cu}\left(C_{4}H_{4}N_{2}\right){\left({\mathrm{NO}}_{3}\right)}_{2}$ and family of HF metals α$-{\mathrm{YbAl}}_{1-x}{\mathrm{Fe}}_{x}B_{4}$, β$-{\mathrm{YbAlB}}_{4}$ and ${\mathrm{YbCo}}_{2}{\mathrm{Ge}}_{4}$. We show that these unexplained experimental data can be explained within the framework of fermion condensation (FC) theory based on the topological FCQPT. As a result, we demonstrate that these HF compounds belong to the new state of matter. Thus, the recent experimental data support our conclusion expressed in recent reviews [13,14] that HF compounds form the new state of matter.

Currently, numerous quantum spin liquids (QSLs) with various types of ground states are proposed [14,15,16,17,18,19,20,21,22,23,24,25]. These QSLs define the thermodynamic, transport and relaxation properties of frustrated insulators and represent the new state of matter formed by HF compounds [1,13,14]. QSLs are formed with fermionic quasiparticles with the effective mass $M^{*}$ which are called spinons. Spinons carry spin σ=1/2 and no charge. At temperature T=0 the Fermi sphere is shaped from spinons with the Fermi momentum $p_{F}$. Thus, frustrated insulators can be viewed as a spinon metal, which differs from HF metals in that it cannot support the electric current.

The Fermi sphere of spinons can be located near the topological Fermion condensation phase transition (FCQPT) that forms the FC state and the corresponding flat band [1,3,4,7,11,26]. In the FC state, at T=0 the corresponding flat band is given by the equation
(1)$$\varepsilon \left(p,T=0\right)=\mu ,\;p_{i}\leq p_{F}\leq p_{f};\;\;0\leq n\left(p\right)\leq 1.$$
where $p_{i}$ and $p_{f}$ stand for initial and final momenta, where the flat band reside. At T>0 the quasiparticle occupation numbers n(p) is given by the Fermi–Dirac distribution function which is represented in the form [5,27]
(2)$$\varepsilon \left(p,T\right)-\mu \left(T\right)=T\ln\frac{1-n(p,T)}{n(p,T)}.$$

Taking into account that T→0, the distribution function satisfies the inequality 0<n(p)<1 for $p_{i}\leq p_{F}\leq p_{f}$, we see that the logarithm is finite; the right hand side of Equation (Equation 2) vanishes, and lead to Equation (Equation 1). Near FCQPT flat band takes place and the notion of the strongly correlated quantum spin liquid (SCQSL) emerges that allows one to describe numerous data related to the thermodynamic, relaxation and transport properties of frustrated magnetic insulators [1,7,11,14,22,28,29,30].

We consider recent measurements in magnetic field *B* of the quantum spin liquid [23] that forms the thermodynamic properties of ${\mathrm{Lu}}_{3}{\mathrm{Cu}}_{2}{\mathrm{Sb}}_{3}O_{14}$ and the thermodynamic of 1DQSL that defines behavior of ${\mathrm{YbAlO}}_{3}$ [24] and $\mathrm{Cu}\left(C_{4}H_{4}N_{2}\right){\left({\mathrm{NO}}_{3}\right)}_{2}$ [22,25], see Section 2. We analyze recently obtained measurements on the heavy-fermion (HF) metals β$-{\mathrm{YbAlB}}_{4}$ [31,32], ${\mathrm{YbCo}}_{2}{\mathrm{Ge}}_{4}$ [33] and α$-{\mathrm{YbAl}}_{1-x}{\mathrm{Fe}}_{x}B_{4}$ [34], that have been performed under the application of magnetic field *B*, as well as on β$-{\mathrm{YbAlB}}_{4}$ under the application of hydrostatic pressure *P* [31,32]. These have received substantial theoretical analysis [31,32,35,36,37,38,39,40,41,42,43,44]. We explain these results within the framework of the FC theory, and show that the mentioned above HF metals exhibit the same behavior as that of SCQSL, forming properties of ${\mathrm{Lu}}_{3}{\mathrm{Cu}}_{2}{\mathrm{Sb}}_{3}O_{14}$, ${\mathrm{YbAlO}}_{3}$ and $\mathrm{Cu}\left(C_{4}H_{4}N_{2}\right){\left({\mathrm{NO}}_{3}\right)}_{2}$, see Section 6. Our results are summarized in Section 8.

## 2. Universal Scaling Behavior of Quantum Spin
Liquid

The ground state energy of QSL depends weakly on the spins configuration, since the spinons of the triangular lattice compounds form symmetric positions. Therefore, the triangular lattice is near to a topologically protected flat band of the spectrum with zero excitation energy [6,7,28,45,46,47]. As a result, the topological FCQPT can be considered as a quantum critical point (QCP) of the ${\mathrm{Lu}}_{3}{\mathrm{Cu}}_{2}{\mathrm{Sb}}_{3}O_{14}$ quantum spin liquid. In that case the elementary magnetic quasiparticles, dubbed spinons, defining the relaxation, transport and thermodynamic properties, carry the effective mass $M^{*}$, zero charge and spin σ=1/2. Spinons occupy the corresponding Fermi sphere with the Fermi momentum $p_{F}$, representing HF quasiparticles of deconfined SCQSL. Spinon quasiparticles generate the excitation spectrum typical for HF compounds located near the topological FCQPT. The ground state energy E(n) is given by the Landau functional, depending on the spinon distribution function $n_{\sigma }\left(p\right)$, where p is the momentum. Near the FCQPT point, the spinon effective mass $M^{*}$ is defined by the Landau Equation [6,48]
(3)$$\begin{array}{ccc} & & \frac{1}{M^{*}\left(T,B\right)}=\frac{1}{M^{*}\left(T=0,B=0\right)}\\ & + & \frac{1}{p_{F}^{2}}\sum_{\sigma_{1}}\int \frac{p_{F}p_{1}}{p_{F}}F_{\sigma ,\sigma_{1}}\left(p_{F},p_{1}\right)\frac{\partial \delta n_{\sigma_{1}}\left(p_{1}\right)}{\partial p_{1}}\frac{dp_{1}}{{\left(2\pi \right)}^{3}}. \end{array}$$

In Equation (Equation 3) *B* is magnetic field and we rewrite the spinon distribution function as $\delta n_{\sigma }\left(p\right)\equiv n_{\sigma }\left(p,T,B\right)-n_{\sigma }\left(p,T=0,B=0\right)$. Note that both functional E(n) and Equation (Equation 3) are exact [1,49]. This fact provides firm ground to construct the theory of HF compounds [1,6,7]. The geometric frustration of QSL in ${\mathrm{Lu}}_{3}{\mathrm{Cu}}_{2}{\mathrm{Sb}}_{3}O_{14}$ [23] is located near the topological FCQPT, therefore we employ the theory of HF compounds, that is the FC theory, to describe SCQSL of ${\mathrm{Lu}}_{3}{\mathrm{Cu}}_{2}{\mathrm{Sb}}_{3}O_{14}$, see, e.g., [6,28]. This theory allows quantitative analysis of the thermodynamic, relaxation and transport properties of both HF compounds containing QSL and HF metals [1,6,7,11,14,28]. We will show that the thermodynamic properties of ${\mathrm{Lu}}_{3}{\mathrm{Cu}}_{2}{\mathrm{Sb}}_{3}O_{14}$ coincide with those of the frustrated magnet ${\mathrm{ZnCu}}_{3}{\left(\mathrm{OH}\right)}_{6}{\mathrm{Cl}}_{2}$ and of HF metals including the archetypical HF metal ${\mathrm{YbRh}}_{2}{\mathrm{Si}}_{2}$ [45].

In Equation (Equation 3) the only role of the Landau interaction is to drive the system to the FCQPT point, where the Fermi surface changes its topology so that the effective mass acquires strong temperature and field dependences [1,6,50,51], as seen from the inset of Figure 1. Indeed, it is seen from the inset that the effective mass $M^{*}\left(T,B\right)\propto C_{\mathrm{mag}}/T$ strongly depends on both *T* and *B*. At the topological FCQPT the term $1/M^{*}\left(T=0,B=0\right)$ vanishes and Equation (Equation 3) becomes homogeneous and therefore is solved analytically. We remark that at $p=p_{F}$ the single-particle spectrum ε(p) acquires an inflection point at which the corresponding flat band related to FC is formed [3,6]. In the simplest case the inflection point turns out to be $\varepsilon \left(p\right)={\left(p-p_{F}\right)}^{3}$ [5,6]; the other types of the inflection points are considered in Section 6. At B=0, the effective mass depends on *T* exhibiting the non-Fermi liquid (NFL) behavior [6]
(4)$$M^{*}\left(T\right)≃a_{T}T^{-2/3}.$$

At finite *T* magnetic field *B* drives the system to the Landau Fermi liquid (LFL) behavior with
(5)$$M^{*}\left(B\right)≃a_{B}B^{-2/3}.$$

Here $a_{T}$ and $a_{B}$ are fitting parameters. Note that the exponent −2/3 corresponds to the above considered inflection point.

**Figure 1 materials-15-03901-f001:**
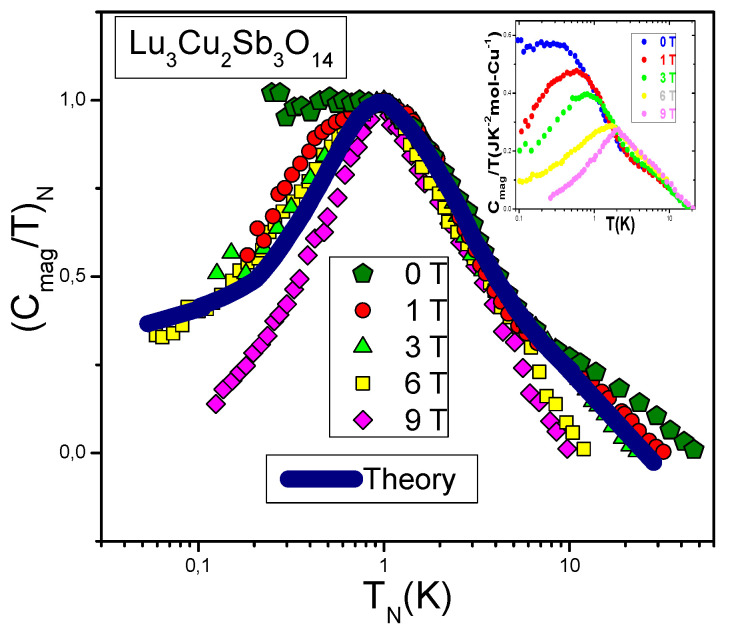
The normalized specific heat ${\left(C_{\mathrm{mag}}/T\right)}_{N}$ as a function of the normalized temperature $T_{N}$ measured under the application of magnetic field is shown in the legend. The normalized specific heat $C_{\mathrm{mag}}/T$ is extracted from the measurement of the specific heat of ${\mathrm{Lu}}_{3}{\mathrm{Cu}}_{2}{\mathrm{Sb}}_{3}O_{14}$ [23] shown in the inset. The solid orange curve displaces the theoretical calculations based on Equation (Equation 3) [45]. The same curve is shown in Figure 2 and Figure 3, exhibiting the scaling of the thermodynamic properties of the quantum spin liquid of ${\mathrm{Lu}}_{3}{\mathrm{Cu}}_{2}{\mathrm{Sb}}_{3}O_{14}$.

The universal scaling of the effective mass $M^{*}$ is shown in Figure 4. This behavior is given by Equation (Equation 3), provided that the system is located near FCQPT. At finite *B* and *T*, near the topological FCQPT, the solution of Equation (Equation 3) $M^{*}\left(B,T\right)$ links the LFL ($M^{*}\left(T\right)\propto const)$ and NFL ($M^{*}\left(T\right)\propto T^{-2/3}$) regions [1,6,7]. As seen from Figure 4, the LFL behavior and the NFL one, given by Equations (Equation 4) and (Equation 5) are separated by the crossover region at which $M^{*}$ reaches its maximum value $M_{M}^{*}$ at temperature $T_{M}$. It is seen from Figure 4, representing the universal scaling behavior of the dimensionless normalized effective mass $M_{N}^{*}=M^{*}/M_{M}$ as a function of the dimensionless normalized temperature $T_{N}=T/T_{M}$, that $M_{N}^{*}$ exhibits the usual behavior of the experimental thermodynamic function like C/T or χ(T), see, e.g., [6]. Indeed, the region $T_{N}∼1$ represents the crossover region between the LFL behavior with almost constant effective mass and the NFL behavior, exhibiting the $M^{*}\propto T^{-2/3}$ dependence, see Equation (Equation 4), and $T_{inf}$ separates the beginning point of the crossover region from the NFL region [6]. Note that both $M_{M}^{*}$ and $T_{M}$, occurring at $M_{N}^{*}=T_{N}=1$, depend on the microscopic properties of the system in question [6], while their normalized values exhibit the universal scaling. Thus, the FC theory, based on the topological FCQPT and the corresponding flat bands, incorporates the inherent universal scaling behavior that is experimentally exhibited by numerous HF compounds [1,6,7].

### 2.1. Universal Behavior of $Lu_{3}Cu_{2}Sb_{3}O_{14}$


It is seen from the inset of Figure 1 that $C_{\mathrm{mag}}/T$ reaches its maximum value ${\left(C_{\mathrm{mag}}/T\right)}_{\max}\left(B\right)$ under the application of magnetic fields at some temperature $T_{\max}\left(B\right)$. To reveal the scaling, we introduce the dimensionless normalized specific heat ${\left(C_{\mathrm{mag}}/T\right)}_{N}$ as a function of the dimensionless normalized temperature $T_{N}=T/T_{\max}\left(B\right)$ [6]
(6)$${\left(C_{\mathrm{mag}}/T\right)}_{N}=\frac{C_{\mathrm{mag}}/T}{{\left(C_{\mathrm{mag}}/T\right)}_{\max}}=M_{N}^{*}.$$

From Equation (Equation 6) and from both Figure 1 and Figure 4, we see that ${\left(C_{\mathrm{mag}}/T\right)}_{N}\left(T_{N}\right)$ exhibits the universal, scaling as a function of only $T_{N}\propto T/B$, presented by a single curve.

To construct the interpolating equation revealing the universal scaling of the effective mass $M^{*}\propto C_{\mathrm{mag}}/T$, we employ both the dimensionless normalized effective mass $M_{N}^{*}$ and the dimensionless normalized temperature $T_{N}$, defined by dividing the effective mass $M^{*}\left(T,B\right)$ by its maximal values, $M_{\max}^{*}\left(T,B\right)$, and temperature *T* by $T_{\max}$ at which the maximum $M_{N}^{*}$ occurs, $T_{N}=T/T_{\max}$ [6]. Magnetic field *B* emerges in Equation (Equation 3) as the combination $\mu_{B}B/k_{B}T$. As a result, $k_{B}T_{\max}≃\mu_{B}B$ where $k_{B}$ is the Boltzmann constant and $\mu_{b}$ is the Bohr magneton [6,50]. This observation allows us to conlcude have that [6,7,44]
(7)$$T_{\max}\propto B$$
and
(8)$$T_{N}\propto T/B$$

Thus, we obtain that the normalized effective mass $M_{N}^{*}=M^{*}/M_{\max}^{*}={\left(C_{\mathrm{mag}}/T\right)}_{N}$ is well defined by the interpolating function, approximating the solutions of Equation (Equation 3) [6]
(9)$$M_{N}^{*}\left(y\right)\approx c_{0}\frac{1+c_{1}y^{2}}{1+c_{2}y^{8/3}}.$$

Here $c_{0}=\left(1+c_{2}\right)/\left(1+c_{1}\right)$, $c_{1}$ and $c_{2}$ are fitting parameters, and $y=T/T_{\max}\propto T/B$. Clearly, from both Equations (Equation 5) and (Equation 9) that under the application of magnetic field $M^{*}$ becomes finite and at low temperatures the system exhibits the LFL behavior, $C_{\mathrm{mag}}\left(T\right)/T\propto M^{*}\left(T\right)≃M^{*}\left(T=0\right)+a_{1}T^{2}$. As seen from the inset of Figure 1, at increasing temperatures $M^{*}\propto C_{\mathrm{mag}}/T$ increases and enters the crossover region, reaching its maximum $M_{\max}^{*}\propto {\left(C_{\mathrm{mag}}\left(T\right)/T\right)}_{\max}$ at $T=T_{\max}$, with subsequent diminishing given by Equations (Equation 4) and (Equation 9). Scaling behavior is manifested by Equation (Equation 9), exhibiting the superior quality of Equation (Equation 3) at the topological FCQPT: the function $M^{*}\left(T,B\right)$ of two variables transforms into the function $M^{*}$ of the single variable $T_{N}\propto T/B$. We employ Equation (Equation 9) to outline the universal scaling, verifying our calculations based on Equation (Equation 3).

The scaling of ${\left(C_{\mathrm{mag}}/T\right)}_{N}=M_{N}$, extracted from the experimental data $C_{\mathrm{mag}}\left(T,B\right)/T$ [23], is reported in Figure 1. The data for a wide range of *B* vales up to 9 T merge well into a single curve. Figure 2 reports the normalized specific heat ${\left(C_{\mathrm{el}}/T\right)}_{N}=M_{N}^{*}$ of ${\mathrm{YbRh}}_{2}{\mathrm{Si}}_{2}$ versus normalized temperature $T_{N}$ as a function of *B*. Indeed, at low $T_{N}≲0.1$ the normalized specific heat ${\left(C_{\mathrm{el}}/T\right)}_{N}≃0.4$ [1,29]. This value is determined by the polarization of the heavy electron band under the application of magnetic fields B>4 T, and coincides with that of $\chi_{N}=M_{N}^{*}$ obtained on ${\mathrm{ZnCu}}_{3}{\left(\mathrm{OH}\right)}_{6}{\mathrm{Cl}}_{2}$ and shown in Figure 3. Results of our calculations are represented by the same solid curve in Figure 1, Figure 2 and Figure 3. We stress that at low normalized temperatures $T_{N}$ and at low magnetic fields *B* the polarization becomes small, making ${\left(C_{\mathrm{el}}/T\right)}_{N}\to 0.9$ at the LFL region, as it is seen from Figure 2. Thus, the scaling behavior of $C_{\mathrm{mag}}/T$ shown in Figure 1 is of universal character; indeed, ${\left(C_{\mathrm{mag}}/T\right)}_{N}=M_{N}^{*}$ of ${\mathrm{Lu}}_{3}{\mathrm{Cu}}_{2}{\mathrm{Sb}}_{3}O_{14}$ behaves like ${\left(C_{\mathrm{el}}/T\right)}_{N}=\chi_{N}=M_{N}^{*}$ shown in Figure 2 and Figure 3 The data shown are extracted from measurements on ${\mathrm{ZnCu}}_{3}{\left(\mathrm{OH}\right)}_{6}{\mathrm{Cl}}_{2}$ and ${\mathrm{YbRh}}_{2}{\mathrm{Si}}_{2}$ [16,52,53]. Thus, the quantum spin liquid of ${\mathrm{Lu}}_{3}{\mathrm{Cu}}_{2}{\mathrm{Sb}}_{3}O_{14}$ can be viewed as SCQSL that exhibits gapless behavior even in the presence of a strong magnetic field. From Figure the data at B=0 and $T_{N}<1$ indicate that the quantum spin liquid exhibits the LFL behavior. Thus, we see that the quantum spin liquid in ${\mathrm{Lu}}_{3}{\mathrm{Cu}}_{2}{\mathrm{Sb}}_{3}O_{14}$ is located before the topological FCQPT. Otherwise, the spin liquid, being on the ordered side of the topological FCQPT, would have been consumed by phase transition, eliminating the corresponding finite value of the residual entropy $S_{0}$ at $S\left(T\to 0\right)\to S_{0}$ [1,6]. In that case one can experimentally observe competition of the different phase transitions at T→0 that make the corresponding phase diagram very complicated, and the only reason of these complexity and competition of possible phase transitions is to vanish $S_{0}$ due to the Nernst law [5,6]. Therefore, we conclude that SCQSL without a gap should be close to the topological FCQPT, and is located on the disordered side of the topological FCQPT.

In Figure 5a, the solid squares denote the values of the maxima ${\left(C_{\mathrm{mag}}/T\right)}_{\max}\left(B\right)$ versus magnetic field *B*, taken from the inset of Figure 1. Clearly the agreement between the theory (solid curve) and the experiment is good. At B=0 the arrow shows the position of the maximum with the subtracted impurity Schottky contribution [23]. We believe that there is no reason to subtract the the impurity Schottky contribution, since it is not possible to differentiate the contribution from the impurities and from those coming from the pure crystal holding SCQSL [45], since both of them form the integral SCQSL [14]. The solid line in Figure 5b represents function $T_{\max}\left(B\right)\propto B$, given by Equation (Equation 8). It is seen that the data are well approximated by the straight line. At B=0$T_{\max}$ is finite, pointing to the fact that SCQSL exhibits the LFL behavior. Indeed, at B=0 and T→0 the specific heat $C_{\mathrm{mag}}/T$ demonstrates the LFL behavior, as evident from the inset of Figure 1. This behavior indicates that SCQSL of ${\mathrm{Lu}}_{3}{\mathrm{Cu}}_{2}{\mathrm{Sb}}_{3}O_{14}$ is placed before the topological FCQPT. Thus, at T→0 the system exhibits the LFL behavior and the absence of a gap or some phase transition, that has to eliminate the residual entropy $S_{0}$. This conclusion is consistent with the general properties of the phase diagrams of HF metals and quantum insulators [14,54]. To clarify the above mentioned properties, one needs to carry out low temperature measurements of both the magnetic susceptibility χ and the thermal transport in presence of the magnetic field.

A few remarks related to the heat transport in frustrated insulators are in order here. Recent measurements of the low-temperature thermal conductivity κ have shown that the value of κ(T→0) strongly depends on the disorder of quantum magnets (insulators) and at high disorder κ(T→0)→0, see, e.g., [55,56]. Measurements on the transition metal dichalcogenide $1T-{\mathrm{TaS}}_{2}$ demonstrates that it hosts QSL on the two-dimensional perfect triangular lattice [56]. Experiments show that the application of magnetic field *B* enhances κ/T and suppresses $C_{mag}/T$ [56]. These observations agree with the predictions of the FC theory [14,44,57]. On the other hand, κ(T→0)→0 could signal that QSL is not present, while the thermodynamic properties of quantum insulators with κ(T→0)→0 demonstrate the typical behavior of HF metals. As a result, we propose that the thermodynamic properties of these insulators are defined by QSL [1,14,28]. Thus, we have to suggest that there are at least two types of QSL: one is represented by QSL with high resistance to the heat transport, that is κ(T→0)→0, and the other is characterized by κ(T→0) being finite. In the latter case κ depends on magnetic field similarly to the magnetoresistance of HF metals [29,30]. We assume that in two dimensional systems, formed by the kagomè lattice, spinons form weakly bound states with impurities and that bound states strongly obstruct the heat transport, κ(T→))→0, but this obstacle does not influence the thermodynamic properties of SCQSL [45].

There are perspective insulators with QSL that are the Kitaev materials. They can be thought of as Mott insulators, exhibiting specific exchange interactions leading to unconventional forms of magnetism induced by QSLs [58]. There are a number of experimentally studied examples like ${\mathrm{Na}}_{2}{\mathrm{IrO}}_{3}$, $\alpha -{\mathrm{Li}}_{2}{\mathrm{IrO}}_{3}$ and α$-{\mathrm{RuCl}}_{3}$ where local magnetic moments are aligned in interacting hexagonal layers, see, e.g., [14,59,60,61]. Measurements of thermal conductivity κ(B) in magnetic fields *B* of the insulator α$-{\mathrm{RuCl}}_{3}$ have shown that κ(B) is finite at T→0 when the antiferromagnetic order is suppressed by magnetic field $B=B_{c}≃7$ T, while κ(B) is an increasing function at $B>B_{c}$ [60,61]. Such a behavior points to the fact that QSL of α$-{\mathrm{RuCl}}_{3}$ is located on the ordered side of the topological FCQPT, and the application of magnetic field shifts the system to the point of FCQPT, similarly to the case of ${\mathrm{YbRh}}_{2}{\mathrm{Si}}_{2}$ [62]. Then, the elevated magnetic field *B* enhances κ, since $\kappa \left(B\right)\propto {\left(M^{*}\left(B\right)\right)}^{-2}$ with $M^{*}\left(B\right)$ given by Equation (Equation 5) [57]. These observations are in good agreement with the behavior of κ(B) of SCQSL, allowing us to suggest that QSL of α$-{\mathrm{RuCl}}_{3}$ represents SCQSL, resembling the corresponding behavior of HF metals [1,14,30,57].

The universal scaling behavior exhibited by ${\mathrm{Lu}}_{3}{\mathrm{Cu}}_{2}{\mathrm{Sb}}_{3}O_{14}$ is shown on Figure 6 that displays T/B scaling of the HF metal ${\mathrm{CeCu}}_{6-x}{\mathrm{Au}}_{x}$ and SCQSL of the frustrated insulator herbertsmithite ${\mathrm{ZnCu}}_{3}{\left(\mathrm{OH}\right)}_{6}{\mathrm{Cl}}_{2}$ [16,63]. This universal scaling demonstrated by the very different HF compounds, allows to conclude that HF compounds represent a new state of matter [1,7]. In contrast to ordinary quantum phase transition, the universal scaling induced by FCQPT occurs up to high temperatures $T<T_{f}$, where $T_{f}∼100$ K, for both LFL and NFL behaviors are defined by quasiparticles (with $M_{N}^{*}$ given by Equation (Equation 9)), rather than by some kind of fluctuations or Kondo lattice [1,6,7]. A few remarks are in order here. A HF compound can be placed before the topological FCQPT, exactly at FCQPT, and behind it on its ordered side, see Section 6, Figure 7. One may expect that the T/B scaling is defined by some phenomena that are not related to both the presence of FCQPT and the corresponding divergence of $M^{*}$, see, e.g., [1,6,7]. On the other hand, if the HF compound in question is located before FCQPT, it exhibits the LFL behavior even without the application of magnetic field *B* at low T→0. At elevated magnetic fields, as magnetic field becomes $B\gg B_{0}$, Equation (Equation 5) is valid and the scaling restores, see also Section 6, Equation (Equation 28). Thus, to observe both scaling and the divergency of the effective mass in measurements on HF compounds, measurements should be performed at sufficiently low temperatures and magnetic fields. For instance, the HF metal ${\mathrm{CeRu}}_{2}{\mathrm{Si}}_{2}$ shows the NFL behavior down to lowest temperatures of 170 mK and very low magnetic fields (B≃0.02 mT) [64]. We note that interpretations of the measurements carried out in presence of magnetic field can lead to incorrect theoretical results that ${\mathrm{CeRu}}_{2}{\mathrm{Si}}_{2}$ demonstrates the LFL behavior at low temperatures [64]. Thus, we have to conclude that a theory is an important tool that allows one to understand what is being measured. For example, a conclusion that the scaling behavior of the thermodynamic properties of a HF compounds without both QCP and the divergency of the effective mass could be caused by simple misinterpretation of the obtained experimental data, see Section 6.

### 2.2. Schematic Phase Diagram of $Lu_{3}Cu_{2}Sb_{3}O_{14}$

Now we are in a position to construct the schematic phase diagram of ${\mathrm{Lu}}_{3}{\mathrm{Cu}}_{2}{\mathrm{Sb}}_{3}O_{14}$ displayed in Figure 8. As seen from (Equation 9) and Figure 1, at T=0 and B=0 the system is located before the topological FCQPT, that is on its disordered side. Therefore, at $T<T_{0}$ the system exhibits the LFL behavior, ensuring the existence of SCQSL without gap, as discussed above. Both magnetic field *B* and temperature *T* play the role of the control parameters, shifting the system from its location at B=0 and $T=T_{0}$ (close to the topological FCQPT) and driving it from the NFL to LFL region as shown by the vertical and horizontal arrows in Figure 8. At a fixed temperature increasing *B* drives the system from the NFL to the LFL region.

This behavior is seen from Figure 5b: $T_{\max}$ increases with *B* increasing. At $T<T_{\max}$ the system exhibits the LFL behavior [6]. On the contrary, at fixed *B* and increasing *T*, the system, following the vertical arrow direction, shifts from the LFL to NFL region. The inset to Figure 8 displays the behavior of the normalized effective mass $M_{N}^{*}$ as a function of the normalized temperature $T_{N}\propto T/B$, as seen from Equation (Equation 9).The region $T_{N}∼1$ represents the crossover region between the LFL behavior with almost constant effective mass and the NFL behavior, exhibiting the $T^{-2/3}$ dependence, see Equation (Equation 4) and inset in Figure 8. We note that in the framework of the Fermion condensation theory it is possible to explain the crossover from the NFL behavior to the LFL one in the presence of the small magnetic fields [1,6,7,14,43], while the applying the pressure does not alter the NFL behavior, provided that the system is located on the ordered side of the topological FCQPT, see, e.g., [32]. In that case the residual entropy $S_{0}$ is eliminated by some phase transition like the superconducting one that occurs in the heavy-fermion superconductor β$-{\mathrm{YbAlB}}_{4}$, later representing a strange metal located away from a magnetic instability, is not accompanied by fluctuations [32,43]. Furthermore, one cannot invoke quantum fluctuations to explain the corresponding properties of the phase diagram Figrue Figure 8 and the dependencies displayed in Figure 5a,b [23]. Thus, the main features of the schematic phase diagram Figrue Figure 8 show that the thermodynamic properties of ${\mathrm{Lu}}_{3}{\mathrm{Cu}}_{2}{\mathrm{Sb}}_{3}O_{14}$ are similar to those of the HF metal ${\mathrm{YbRh}}_{2}{\mathrm{Si}}_{2}$ and ${\mathrm{ZnCu}}_{3}{\left(\mathrm{OH}\right)}_{6}{\mathrm{Cl}}_{2}$ [1,6,14]. As a result, the quantum spin liquid of ${\mathrm{Lu}}_{3}{\mathrm{Cu}}_{2}{\mathrm{Sb}}_{3}O_{14}$ is represented by SCQSL [45].

## 3. Quasi-One Dimensional Quantum Spin Liquids

The behavior of quasi-one dimensional quantum spin liquid (1DQSL) is the subject of ongoing intensive experimental research in condensed matter physics (see, e.g., [22,24,25] and references therein). Recently, searching for 1DQSL, the salient experiments were performed on the 1D Heisenberg antiferromagnet insulators Cu(C${}_{4}$H${}_{4}$N${}_{2}$)(NO${}_{3}$)${}_{2}$ (CuPzN) and ${\mathrm{YbAlO}}_{3}$ in the presence of magnetic field and interpreted in terms of SCQSL and the Tomonaga-Luttinger liquid (TLL) [22,24,25]. The observed thermodynamic properties of both CuPzN and ${\mathrm{YbAlO}}_{3}$ are atypical and it is expected that they do not belong to the class of HF compounds, including HF metals and quasicrystals insulators with quantum spin liquid and HF metals [6,7,12,22,28,65]. In this Section we show that, contrary to conventional wisdom, both CuPzN and ${\mathrm{YbAlO}}_{3}$ can be considered as insulators belonging to HF compounds, while their thermodynamic properties are defined by weakly interacting 1DQSL formed by spinons, and are similar to those of the HF compounds.

One dimensional (1D) chain of half-odd-integer spins described by the Heisenberg model can be mapped on the fermionic system [66,67,68,69]. One of the hallmark features of geometrically frustrated insulators is the spin-charge separation; frustrated spin system is disconnected from the electron system characterized by the charge gap, and forms approximate flat band, as an electron system does in metals, e.g., formed by Na [1,14]. At T=0 1DQSL survives up to the saturation field $B_{s}∼2J$, with *J* being the exchange coupling constant, e.g., between Cu${}^{2+}$ in the 1D chains [25]. At $B=B_{s}$ the QCP occurs, creating the gapped field-induced paramagnetic phase [25,67,70]. That is, at $B\to B_{s}$ both antiferromagnetic (AFM) sublattices align toward the field direction, and the magnetic field $B_{s}$ fully polarizes 1DQSL spins, forming flat bands [22]. Thus, in 1DQSL the topological FCQPT plays a role of QCP, at which the energy band for spinons becomes almost flat at $B=B_{s}$ due to the purely kinematic reasons, and the effective mass $M^{*}$ of spinons diverges [22]. Beyond FCQPT at $B>B_{s}$ 1DQSL is fully polarized, leading to the vanishing magnetic susceptibility χ(T→0). Therefore, CuPzN and ${\mathrm{YbAlO}}_{3}$ can be considered as weakly interacting fermions with simplest possible spectrum $\varepsilon =p^{2}/\left(2m_{0}\right)$, where *p* is the momentum and $m_{0}$ is the bare mass (we use the atomic units ℏ=c= 1). In vicinity of FCQPT occurring at $B=B_{s}$ and T=0, the fermion spectrum becomes almost flat, since at FCQPT $p_{F}\to 0$. The spinon effective mass diverges, $M^{*}\propto m_{0}/p_{F}\to \infty$, as it is seen from Figure 9 [22]. In case of weak repulsion between spinons the divergence is associated with the onset of a topological transition at finite value of $p_{F}$ signaling that $M^{*}\left(T\right)\propto T^{-1/2}$, see Section 6 and Refs. [12,22,71,72,73]. Following [74], we assume that the weakly interacting 1DQSL could be thought as QSL formed by fermionic spinons generating the Fermi sphere (line) with the Fermi finite momentum $p_{F}$, and carrying spin 1/2 and no charge. This observation is supported by experimental facts collected on quasi-1D HF metal ${\mathrm{YbNi}}_{4}P_{2}$ [75]. These facts demonstrate that the spin-charge separation is not observed, while the thermodynamic properties of ${\mathrm{YbNi}}_{4}P_{2}$ resemble those of HF metals including the emergence of the LFL behavior under the application of magnetic field with the resistance $\rho \left(T\right)\propto T^{2}$ [75]. Note that recently a new state of matter, quasi-Fermi liquid, has been introduced [76,77] in context of 1DQSL with the bare interaction of spinons being weak. In that case the original Tomonaga-Luttinger system can exactly be mapped on a system of free spinons, whose low-temperature behavior in magnetic fields exhibits the LFL one [74]. As a result, we shall see that the T−B phase diagram of 1DQSL in both CuPzN and ${\mathrm{YbAlO}}_{3}$ resembles that of HF compounds. Thus, CuPzN, ${\mathrm{YbAlO}}_{3}$ and ${\mathrm{YbNi}}_{4}P_{2}$ represent a unique possibility to observe a new type of 1DQSL whose thermodynamic properties resemble that of HF compounds including HF metals.

As 3D AFM ordering is observed in CuPzN and ${\mathrm{YbAlO}}_{3}$ [24,78] with very weak coupling $J^{\prime }$ between Cu${}^{2+}$ chains, the proper spin Hamiltonian is of the form
(10)$$B=J\sum_{i}S_{i}\cdot S_{i+1}+J^{\prime }\sum_{<ij>}S_{i}\cdot S_{j}-B\sum_{i}S_{iz},$$
with *J* being the interchain exchange coupling constant and $J^{\prime }<<J$ is the interchain coupling. The experimental values J≈ 10.3 K [79] and $J^{\prime }\approx$ 0.046 K [78] allow to conclude that the criterion of smallness of the ratio is met, $J^{\prime }/J\approx$ 0.0045. The Holstein-Primakoff model of the bozonization of the spin Hamiltonian (Equation 10) up to first nonlinear terms shows that their contribution to magnetization turns to be about 5% of that stemming from noninteracting boson gas [66,68,80,81,82].

As a result, both CuPzN and ${\mathrm{YbAlO}}_{3}$ are indeed represented by weakly interacting fermions. Therefore, magnetization in terms of fermion number per spin is given by
(11)$$N/L=\int_{0}^{\infty }D\left(\varepsilon \right)f\left(\varepsilon -\mu \left(B\right)\right)d\varepsilon .$$

Here *L* is the number of spins in 1D chain, D(ε) is the density of states, corresponding to free fermion spectrum $\varepsilon =p^{2}/\left(2m_{0}\right)$. The chemical potential $\mu \left(B\right)=B_{s}-B$, and $f\left(x\right)={\left(e^{x}+1\right)}^{-1}$ is the well-known Fermi distribution function [25,68,69,83,84,85]. The magnetization can be expressed as $M=M_{s}-N$ ($M_{s}$ is the saturation magnetization)
(12)$$M\left(B,T\right)=M_{s}-\frac{\sqrt{2m_{0}T}}{\pi }\int_{0}^{\infty }\frac{dx}{e^{\left(x^{2}-\frac{B_{s}-B}{T}\right)}+1}.$$

Equations (Equation 9) and (Equation 12) will be used below to calculate the differential magnetic susceptibility χ
(13)$$\chi \left(T,B\right)=\frac{\partial M(T,B)}{\partial B}$$

Dimensionless normalized magnetic susceptibility $\chi_{N}$ of 1DQSL versus dimensionless variable $(T/|B-B_{s}{\left|\right)}_{N}$ for magnetic field below and above the saturation field $B_{s}$ is displayed in Figure 10. In the fermion representation of the 1DQSL ground state energy E(n), it can be viewed as the Landau functional depending on the spinon distribution function $n_{\sigma }\left(p\right)$, where p is the momentum. Near the topological FCQPT point, the effective mass $M^{*}$ is governed by the Landau Equation (Equation 3). In that case the main role of the Landau interaction $F\left(p_{1},p_{2}\right)=\delta^{2}E/\delta n\left(p_{1}\right)\delta n\left(p_{2}\right)$ is to bring the system to the topological FCQPT point, where $M^{*}\to \infty$ at T=0, and the Fermi surface alters its topology so that the effective mass acquires the temperature and the magnetic field dependences, while the proportionality of the specific heat C/T and the magnetic susceptibility χ to $M^{*}$ holds: $C/T∼\chi ∼M^{*}\left(T,B\right)$ [6,7,50,51]. This feature can be used to separate solutions of the Equation (Equation 3), corresponding to specific experimental situation, for details see Section 6. Namely, Namely, experiments on ${\mathrm{YbAlO}}_{3}$ [24] and on $\mathrm{Cu}\left(C_{4}H_{4}N_{2}\right){\left({\mathrm{NO}}_{3}\right)}_{2}$ [22,25] show that near the topological FCQPT at $B=B_{s}$ the temperature $T_{M}$ at which the maximum value of χ occurs vanishes, $T_{M}\to 0$, see Figure 11a,b. In accordance with these observations, the magnetic susceptibility of CuPzN diverges as $\chi \left(T\right)\propto T^{-1/2}$ at B=13.55 T [25] meaning that the divergence of $M^{*}$ is responsible for the observed behavior, as seen from Figure 9. Again, we recognize QCP at $B=B_{s}$ as the topological FCQPT.

It has been shown that near the FCQPT effective mass can behave as $M^{*}\left(T\right)\propto T^{-1/2}$, see Section 6, while the application of *B* drives the system to the LFL region with $M^{*}\left(B\right)\propto {\left(B_{s}-B\right)}^{-1/2}$ [7,12]. At finite *B* and *T*, near FCQPT, solutions of Equation (Equation 3) $M^{*}\left(T,B\right)$ can be well approximated by a simple universal interpolating function [6,7,12]. The interpolation occurs between the LFL ($M^{*}\propto a+bT^{2}$) and NFL ($M^{*}\propto T^{-1/2}$) regimes and represents the universal scaling of $M_{N}^{*}\left(T_{N}\right)$ independent of the spatial dimension of the considered system
(14)$$M_{N}^{*}=\frac{1+c_{2}}{1+c_{1}}\frac{1+c_{1}T_{N}^{2}}{1+c_{2}T_{N}^{5/2}},$$
where $c_{1}$ and $c_{2}$ are fitting parameters, $M_{N}^{*}=M/M_{M}^{*}$ and $T_{N}=T/T_{M}$ are the normalized effective mass and temperature respectively. Here, [6,7,12]
(15)$$M_{M}^{*}\propto {\left|B_{s}-B\right|}^{-1/2},$$
(16)$$T_{M}\propto \left|B_{s}-B\right|.$$

We remaind that $M_{M}^{*}$ is the maximum value of the effective mass $M^{*}$ taking place at $T_{M}$. It is seen from Equation (Equation 16) that the normalized temperature reads
(17)$$T_{N}\propto T/\left|B_{s}-B\right|$$

As a result, we obtain that $\chi \left(T,B\right)/\chi_{\max}\left(T,B\right)=M_{N}^{*}\left(T_{N}\right)$ becomes a function of the single variable $T_{N}\propto T/B$ as it is shown in Figure 4, see, e.g., [1,6,7]. Here $\chi_{\max}\left(T,B\right)$ is the maximum of χ(T,B) occurring at $T_{M}\left(B\right)$. Below Equations (Equation 14) and (Equation 17) is used along with Equation (Equation 12) to describe the experimental facts collected on CuPzN and ${\mathrm{YbAlO}}_{3}$.

## 4. Experiment versus Theory

As seen from Figure 12a,b our calculations of $\chi_{N}$ for CuPzN and ${\mathrm{YbAlO}}_{3}$ represented by solid curves agree quite well with the experimental values from [24,25]. Magnetic susceptibility at $B<B_{s}$ exhibits the LFL behavior in magnetic fields at which $T/(B_{s}-B)<1$, and at rising temperatures, that is $T/(B_{s}-B)>1$, the NFL behavior occurs. In between of the LFL and NFL behavior, the crossover takes place with the maximum of $\chi_{N}\left(T/B\right)$.

Theory agrees similarly well with the experiment in case $B>B_{s}$, as shown in Figure 13a,b. Now, since he system is fully polarized, $\chi_{N}\to 0$ at $T/(B_{s}-B)\to 0$; at temperature increasing polarization dissolves, leading to increasing $\chi_{N}$. Then the $\chi_{N}$ reaches its maximum value and the growing is intercepted as soon as the NFL behavior sets in.

The comparison of the experimental results for the normalized magnetization ${\left(M_{c}/\sqrt{B}\right)}_{N}$ obtained on β$-{\mathrm{YbAlB}}_{4}$ and CuPzN show very good agreement between these very different compounds, as seen from Figure 14. This result is in a good agreement with the theoretical curve taken from [22,43], as seen from Figure 14 that reports the scaling behavior of the magnetization $M_{c}/B^{0.5}=a+\left(M-M_{s}\right)/{\left(B_{s}-B\right)}^{0.5}$ as a function of $T/B=T/(B_{s}-B)$, with *a* being a constant. Indeed, from Figure 14, the LFL behavior occurs at T≪B, the crossover at T∼B, and the NFL one at T≫B, as in the case of the HF compounds [6,7,43]. As a result, we conclude that HF metals and 1DQSL exhibit the behavior similar to that of β$-{\mathrm{YbAlB}}_{4}$. In Section 6, we show that the HF metals β$-{\mathrm{YbAlB}}_{4}$ exhibits the same universal behavior that α$-{\mathrm{YbAl}}_{1-x}{\mathrm{Fe}}_{x}B_{4}$ and ${\mathrm{YbCo}}_{2}{\mathrm{Ge}}_{4}$ do.

## 5. Phase Diagram of One Dimensional Quantum Spin Liquids

Thermodynamic properties reported in Figure 10, Figure 12 and Figure 13 allow us to construct the T−B phase diagram of 1DQSL, shown in Figure 15. We see from Figure 10, that the peak temperature $T_{M}$ vanishes as *B* approaches $B_{s}$. Furthermore, from Figure 9 the effective mass $M^{*}$ of spinons does diverge at $B\to B_{s}$, since it takes place at FCQPT. Based on these observations we construct the T−B phase diagram reported in Figure 15, demonstrating that the peak dependence $T_{M}$ takes place over the wide range of *B*, for $T_{M}\propto \left(B_{s}-B\right)$. Thus, we conclude that the curves $T_{M}\left(B\right)$ are straight lines, representing energy scales typical for HF metals located at their QCP [1,54,86]. Since FCQPT occurs at $B=B_{s}$, the phase diagram is approximately symmetric with respect to the point $B=B_{s}$, and consists of the LFL, gapped Fermi liquid, crossover and the NFL regions. Some asymmetry comes from the impediment that the LFL region may be occupied by some ordered phases marked by OP in Figure 15, as it happens for ${\mathrm{YbAlO}}_{3}$ [24]. The crossover regions in Figure 15 are depicted by arrows, and are represented by the straight lines that represent the *B*-dependencies of temperatures of approximate LFL and NFL boundaries as well as by that of $T_{M}$. It is seen that NFL state occurs at relatively high temperatures. At the same time LFL region are located at low temperatures, where the spinon effective mass $M^{*}$ is almost constant, characteristic to the LFL behavior. At $B>B_{s}$ the 1DQSL transforms into a gapped field-induced paramagnetic spin liquid, as shown in Figure 15. With temperature increasing and the fixed magnetic field *B*, 1DQSL transits through the crossover, and enters the NFL region. The crossover region shown by oliver circles becomes wider, as 1DQSL moves from the topological FCQPT displayed by the filled red circle. We conclude that 1DQSL exhibits the typical behavior of HF compounds [1,54] forming the corresponding T−B phase diagram displayed in Figure 15.

## 6. Universal Scaling in Heavy Fermion
Metals

To address the universal scaling within the context of the topological FCQPT, we begin with examination of the scaling of the thermodynamic functions of α$-{\mathrm{YbAl}}_{1-x}{\mathrm{Fe}}_{x}B_{4}$. The Landau functional E(n) representing the ground-state energy depends on the quasiparticle momentum distribution $n_{\sigma }\left(p\right)$. Near the topological FCQPT, the effective mass $m^{*}$ is governed by the Landau equation Equation (Equation 3). In this section, in order reserve the capital letter for magnetization *M*, we denote the effective mass by $m^{*}$. Let us recall that the Landau functional E(n) and Equation (Equation 3) are exact expressions [6,49]. The Landau interaction $F\left(p_{1},p_{2}\right)=\delta^{2}E/\delta n\left(p_{1}\right)\delta n\left(p_{2}\right)$ brings the system to the FCQPT point when $m^{*}\to \infty$ at T=0. At this point the topology of the Fermi surface is altered, and in contrast to the LFL theory, near this point the effective mass $m^{*}$ acquires strong temperature and field dependencies. However, the typical LFL theory relations
(18)$$C/T∼\chi ∼m^{*},$$
remain intact. Approaching the FCQPT, $m^{*}\left(T=0,B=0\right)\to \infty$ and Equation (Equation 3) becomes homogeneous, i.e., $m^{*}\left(T=0,B\right)\propto B^{-z}$ and $m^{*}\left(T,B=0\right)\propto T^{-z}$, with *z* depending on the analytical properties of *F* [5,6,7,12]. On the ordered side of FCQPT at T=0, the single-particle spectrum ε(p) becomes flat in some interval $p_{i}<p_{F}<p_{f}$ surrounding the Fermi surface at $p_{F}$. Thus, under the influence of the topological FCQPT, the two dimensional (2D) Fermi surface transforms into 3D Fermi volume:(19)ε(p)=μ,
where μ is the chemical potential. At FCQPT the flat interval shrinks, since $p_{i}\to p_{F}\to p_{f}$, and ε(p) possesses an inflection point at $p_{F}$, with $\varepsilon \left(p≃p_{F}\right)-\mu ≃{\left(p-p_{F}\right)}^{3}$. This inflection point can also emerge in the case of a non-analytical Landau interaction *F*, with [43]
(20)$$\begin{array}{ccc} \varepsilon (p)-\mu & ≃ & -{\left(p_{F}-p\right)}^{2},p<p_{F}\\ \varepsilon (p)-\mu & ≃ & \;\;\;{\left(p-p_{F}\right)}^{2},p>p_{F}. \end{array}$$

At the inflection point given by Equation (Equation 20) the effective mass diverges as $m^{*}\left(T\to 0\right)\propto T^{-1/2}$ [1,12,43]. These features of ε(p) can be used to specify the solutions of Equation (Equation 3) according to different experimental situations. In particular, the experimental results obtained for both $\beta -{\mathrm{YbAlB}}_{4}$ and α$-{\mathrm{YbAl}}_{1-x}{\mathrm{Fe}}_{x}B_{4}$ show that near QCP at B≃0, the magnetization obeys $M\left(B\right)\propto B^{-1/2}$ [12,31,35,36,37,42]. This behavior corresponds to the spectrum ε(p) given by Equation (Equation 20) with $\left(p_{f}-p_{i}\right)/p_{F}\ll 1$. Near the FCQPT and at finite *B* and *T*, the solutions of Equation (Equation 3) determining the *T* and *B* dependencies of $m^{*}\left(T,B\right)$ can be well approximated by the universal interpolating function [1,6,7,12]. The interpolation used between the LFL regime ($m^{*}\propto a+bT^{2}$) with
(21)$$m^{*}\propto B^{-1/2},$$
and the NFL is given by Equation (Equation 4).

The regimes given by Equations (Equation 4) and (Equation 15) are separated by the crossover region at which $m^{*}$ reaches its maximum value $m_{M}^{*}$ at temperature $T_{M}$, as seen from Figure 4, representing the universal scaling of the dimensionless normalized effective mass $m_{N}^{*}=m^{*}/m_{M}$ as a function of the dimensionless normalized temperature $T_{N}=T/T_{M}$. Evidently, the region $T_{N}∼1$ represents the crossover region between the LFL behavior with almost constant effective mass and the NFL behavior, exhibiting the $m^{*}\propto T^{-1/2}$ dependence, see Equation (Equation 4). As we shall see below, the inflection point $T_{inf}$ can be used to reveal the universal scaling behavior. In Figure 4, $T_{inf}$ shows approximately the beginning point of the crossover region [6]. Note that both $m_{M}^{*}$ and $T_{M}$ depend on the microscopic properties of the system in question [6], while the normalized values exhibit universal scaling given by the equation:(22)$$m_{N}^{*}\left(T_{N}\right)=\frac{m^{*}\left(T,B\right)}{m_{M}^{*}}=\frac{1+c_{2}}{1+c_{1}}\frac{1+c_{1}T_{N}^{2}}{1+c_{2}T_{N}^{5/2}}.$$

Here $c_{1}$ and $c_{2}$ are fitting parameters. Clearly, from Equations (Equation 4) and (Equation 15), see, e.g., [1,44],
(23)$$T_{M}\propto T_{inf}\propto B\;\;;\;\;T_{N}=T_{M}/T\propto T/T_{inf}\propto T/B.$$

From Equations (Equation 14) and (Equation 16) $m^{*}$ is seen to exhibit the universal scaling [1,6,44]
(24)$$m^{*}\left(T,B\right)=c_{3}\frac{1}{\sqrt{B}}m_{N}^{*}\left(T/B\right),$$
where $c_{3}$ is a constant [6,7,12]. We describe the experimental observations on α$-{\mathrm{YbAl}}_{1-x}{\mathrm{Fe}}_{x}B_{4}$ using Equations (Equation 14) and (Equation 24). Note that the scaling occurs at temperatures $T≲T_{f}$, with $T_{f}$ being the temperature at which the influence of the FCQPT becomes negligible [6,7]. Based on Equation (Equation 24), we conclude that magnetization *M* as described within the theory of fermion condensation does exhibit the empirical scaling behavior, given by
(25)$$\frac{dM(T,B)}{dT}=\int \frac{d\chi (y)}{dT}dB_{1}\propto -\frac{1}{\sqrt{B}}\int \frac{dm_{N}^{*}\left(y\right)}{dy}\frac{dy}{y},$$
where y=T/B. Indeed, as seen from Equation (Equation 25), $\sqrt{B}dM/dT$ is a function of the only variable *y*.

To confirm the validity of Equation (Equation 25) and to demonstrate the universal scaling, we show in Figure 16 our calculated dimensionless normalized magnetization ${\left(B^{1/2}dM\left(T,B\right)/dT\right)}_{N}$ versus the dimensionless normalized ratio ${\left(T/B\right)}_{N}$. The normalization is obtained by dividing $B^{1/2}dM\left(T,B\right)/dT$ and T/B by their maximum values ${\left(B^{1/2}dM\left(T,B\right)/dT\right)}_{M}$ and ${\left(T/B\right)}_{M}$ respectively. We recall that it is the normalization that reveals the universal behavior of HF compounds, since it allows one to get rid of microscopic properties of HF compounds, thereby elucidating their universal properties [1,6].

Evidently, as it is seen from Figure 16, the calculated scaling function of the ratio ${\left(T/B\right)}_{N}$, taken from [43], tracks the data of the normalized quantity ${\left(B^{1/2}dM\left(T,B\right)/dT\right)}_{N}$ well. It also follows from Equation (Equation 25) that the calculated function ${\left(B^{1/2}dM\left(T,B\right)/dT\right)}_{N}$ exhibits the scaling as a function of ${\left(B/T\right)}_{N}$. The NFL, crossover and LFL behavior are indicated in Figure 16a–c by arrows. The theory is represented by the solid curve [43], describing very well the scaling of ${\left(B^{1/2}dM\left(T,B\right)/dT\right)}_{N}$ for the HF metals β$-{\mathrm{YbAlB}}_{4}$, α$-{\mathrm{YbAl}}_{1-x}{\mathrm{Fe}}_{x}B_{4}$ and ${\mathrm{YbCo}}_{2}{\mathrm{Ge}}_{4}$. It is evident from Figure 16 that our calculations, not involving fitting parameters and ad hoc functions, are in good agreement with the experimental data [32,33,34,37,38].

## 7. Schematic Temperature—Doping Phase Diagram

To construct the schematic phase diagram of α$-{\mathrm{YbAl}}_{1-x}{\mathrm{Fe}}_{x}B_{4}$, we define the location of α$-{\mathrm{YbAl}}_{1-x}{\mathrm{Fe}}_{x}B_{4}$ with respect to the topological FCQPT. The locations of the HF metals β$-{\mathrm{YbAlB}}_{4}$ and ${\mathrm{YbCo}}_{2}{\mathrm{Ge}}_{4}$ are defined: β$-{\mathrm{YbAlB}}_{4}$ is located beyond FCQPT, and ${\mathrm{YbCo}}_{2}{\mathrm{Ge}}_{4}$ before FCQPT [43,44]. When applying magnetic field *B* and at sufficiently low temperatures, α$-{\mathrm{YbAl}}_{1-x}{\mathrm{Fe}}_{x}B_{4}$ is driven to the LFL state having with the resistivity [34]
(26)$$\rho \left(T\right)=\rho_{0}+A\left(B\right)T^{2}.$$

Measurements of the coefficient A(B) provides information on the location of the corresponding HF metal with respect to the topological FCQPT. Being proportional to the quasiparticle-quasiparticle scattering cross section, A(B) obeys the relation $A\propto {\left(m^{*}\left(B\right)\right)}^{2}$ [6,62,87], provided that system is located at the point of FCQPT. According to Equation (Equation 15) with $B_{s}=0$, this implies that
(27)$$A\left(B\right)\propto \frac{1}{B}.$$

If the system is located before FCQPT, then at low temperatures and as B→0, the coefficient A(B) acquires the LFL behavior and can be approximated by the interpolating function, see, e.g., [6,44]
(28)$$A\left(B\right)=\frac{a_{1}}{\sqrt{B^{2}+a_{2}}},$$
where $a_{1}$ and $a_{2}$ are fitting parameters. From Equation (Equation 28), as B→0 the coefficient A→const, similarly to the case of LFL behavior; at elevated magnetic field *B* one observes the behavior given by Equation (Equation 27). Figure 17 presents the fit of A(B) to the data extracted from the experimental data [34].

It is seen from Figure 17, that the theoretical dependence (Equation 28) agrees very well with the experimental data leading to a conclusion that the physics underlying the field-induced re-entrance into the LFL behavior under the application of magnetic field *B* is the same as for the HF metals and is defined by Equations (Equation 22) and (Equation 27). It is important to note here that deviations of the theoretical curve at low values of *B* from that given by Equation (Equation 27) are due to the fact that $\alpha -{\mathrm{YbAl}}_{1-x}{\mathrm{Fe}}_{x}B_{4}$ is located before the topological FCQPT, and since now the system exhibits the LFL behavior, $m^{*}$ does not diverge. To confirm this conclusion, we analyze the behavior of the resistivity $\rho_{a}\left(T\right)$ at low temperatures, as shown in Figure 18a,b.

From Figure 18a it is clearly seen that for 1<T<5 K the resistivity demonstrates the typical NFL behavior characterized by the linear dependence $\rho_{a}\left(T\right)\propto T$. At low temperatures T→0 this behavior is violated. Figure 18b shows that as T→0 the violation is defined by the LFL behavior, since the *T*-dependence of the resistivity is given by Equation (Equation 26). This observed agreement is consistent with that derived above from Figure 17.

We now turn to Figure 19 that displays the magnetization M(T) as a function of *T* for different values of the magnetic field *B*. In Figure 5, the inset displays the inflection point $T_{inf}$ versus magnetic field *B*, and shows that at B=0 the inflection point has a finite value $T_{inf}≃3$ mK.

From Equation (Equation 18) it follows that at the crossover region, starting at $T_{inf}$, magnetization behaves as $M/B\propto \chi \propto m^{*}\left(T/B\right)$, as seen from Figure 4. Since from Equation (Equation 23), $T_{inf}\propto B$, the finite value of $T_{inf}$ as B→0 signals that the system is located before the topological FCQPT, exhibiting the LFL behavior at B=0 and T→0. This observation is consistent with the LFL behavior of both A(B→0) shown in Figure 17 and the resistivity shown in Figure 18b.

Based on the results of the above analysis, we can construct the schematic $T-x/x_{c}$ phase diagram of α$-{\mathrm{YbAl}}_{1-x}{\mathrm{Fe}}_{x}B_{4}$ with $x_{c}=0.014$ depicted in Figure 7, with the doping $x/x_{c}$ as a control parameter. At T→0, B=0 and $x/x_{c}>1$ the system exhibits the LFL behavior, and, therefore, is located before the topological FCQPT, that is on its disordered side, as it is shown by the blue arrow. Thus, at T→0 the system exhibits the LFL behavior, and we expect that the scaling of α$-{\mathrm{YbAl}}_{1-x}{\mathrm{Fe}}_{x}B_{4}$ (with $x_{c}=0.042$ and $x_{c}=0.014$) to be violated at low temperatures. At increasing temperatures *T* and fixed magnetic field *B* the system enters the crossover region and continues into the NFL one, displaying the restoration of the scaling behavior. We suggest that a fine tuning of *x* can place α$-{\mathrm{YbAl}}_{1-x}{\mathrm{Fe}}_{x}B_{4}$ at FCQPT with the T/B scaling down to lowest temperatures, while the doping x=0.042 drives the system from FCQPT, and the system still exhibits the LFL behavior at relatively high temperatures [34]. Now we employ the phase diagram Figure 8 to demonstrate the similarity between frustrated insulators and HF metals. As seen from Figure 8, both the magnetic field *B* and the temperature *T* play the role of the control parameters, shifting the system from its location close to the topological FCQPT and driving it from the NFL to LFL regions as shown by the vertical and horizontal arrows. At fixed temperatures $T>T_{0}$ increasing *B* drives the system from the NFL to the LFL region. On the contrary, at fixed *B* and growing temperatures *T*, the system goes along the vertical arrow from the LFL to the NFL region. The same behavior is exhibited by the HF metal ${\mathrm{YbCo}}_{2}{\mathrm{Ge}}_{4}$ that does not demonstrate scaling down to lowest temperatures, since at $x/x_{c}$ it is located before the quantum critical point associated with the topological FCQPT, see Figure 7. For the same reason the effective mass does not diverge at the lowest temperatures [44]. It is worth noting that within the framework of the theory of fermion condensation it is possible to explain the crossover from the NFL behavior to LFL when the small magnetic field is applied [1,6,43]. However, application of pressure does not change the NFL behavior, see, e.g., [32]. Such a behavior observed in the heavy-fermion superconductor β$-{\mathrm{YbAlB}}_{4}$, a strange metal located away from a magnetic instability, is not accompanied by fluctuations [32]. Therefore, at B=0, β$-{\mathrm{YbAlB}}_{4}$ acquires a flat band, implying the presence of a fermion condensate in a strongly degenerate state of matter that becomes susceptible to transition into a superconducting state [43]. Accordingly, the HF metal β$-{\mathrm{YbAlB}}_{4}$ is located behind FCQPT. Thus, the general features of the schematic phase diagram Figure 8 demonstrate that at elevated T/B the thermodynamic properties of α$-{\mathrm{YbAl}}_{1-x}{\mathrm{Fe}}_{x}B_{4}$ become close to those of the HF metals ${\mathrm{YbCo}}_{2}{\mathrm{Ge}}_{4}$ [44] and β$-{\mathrm{YbAlB}}_{4}$ [43], as it is seen from Figure 16. Note that at the doping x=0.014 and relatively high temperatures [34] and elevated magnetic field *B* the system enters the LFL region from the transition region, and the behavior $\rho \left(T\right)\propto T^{2}$ emerges, see, e.g., [6,7]. To carefully locate the position of the HF metal α$-{\mathrm{YbAl}}_{1-x}{\mathrm{Fe}}_{x}B_{4}$ with respect to the topological FCQPT one needs to carry out measurements of the resistivity at low temperatures at which the system is placed at the LFL region, as shown by the blue arrow in Figure 7. We suggest that this procedure can allow to tune the HF metal α$-{\mathrm{YbAl}}_{1-x}{\mathrm{Fe}}_{x}B_{4}$ to the topological FCQPT point by doping *x*.

## 8. Summary

We have demonstrated that the quantum spin liquid of ${\mathrm{Lu}}_{3}{\mathrm{Cu}}_{2}{\mathrm{Sb}}_{3}O_{14}$ and quasi-one dimensional quantum spin liquid of both ${\mathrm{YbAlO}}_{3}$ and $\mathrm{Cu}\left(C_{4}H_{4}N_{2}\right){\left({\mathrm{NO}}_{3}\right)}_{2}$ can be considered as strongly correlated Fermi systems whose thermodynamic properties are defined by SCQSL located near FCQPT. Our calculations of thermodynamic properties and the constructed phase diagrams are in a good agreement with the experimental data. Thus, the quantum spin liquid and 1DQSL are well represented by SCQSL and well described within the theory of fermion condensation [1,6,7,14,45]. We remark that the observed universal scaling can hardly be explained within theories based on some kinds of fluctuations.

We have also analyzed the thermodynamic and transport properties of the heavy-fermion metal α$-{\mathrm{YbAl}}_{1-x}{\mathrm{Fe}}_{x}B_{4}$ and explained its enigmatic scaling behavior within a topological description based on the topological FCQPT. The similarity between the three different HF metals α$-{\mathrm{YbAl}}_{1-x}{\mathrm{Fe}}_{x}B_{4}$, ${\mathrm{YbCo}}_{2}{\mathrm{Ge}}_{4}$ and β$-{\mathrm{YbAlB}}_{4}$ has been explained, and have shown that their T/B scaling can be described by the same universal function. We predict that a fine tuning of *x* around $x_{c}=0.014$ can place α$-{\mathrm{YbAl}}_{1-x}{\mathrm{Fe}}_{x}B_{4}$ at FCQPT with the T/B scaling down to lowest temperatures, namely $T_{0}\to 0$, with the effective mass exhibiting divergent behavior $m^{*}\left(T\right)\propto T^{-1/2}$ down to lowest temperatures. We have also demonstrated that the fermion condensation theory provides good description of the scaling for various HF compounds. Our results are in a good agreement with experimental observations and allow us to conclude that both the HF metals α$-{\mathrm{YbAl}}_{1-x}{\mathrm{Fe}}_{x}B_{4}$, β$-{\mathrm{YbAlB}}_{4}$ and ${\mathrm{YbCo}}_{2}{\mathrm{Ge}}_{4}$ and the frustrated insulators ${\mathrm{Lu}}_{3}{\mathrm{Cu}}_{2}{\mathrm{Sb}}_{3}O_{14}$, $\mathrm{Cu}\left(C_{4}H_{4}N_{2}\right){\left({\mathrm{NO}}_{3}\right)}_{2}$ and ${\mathrm{YbAlO}}_{3}$ with SCQSL form the new state of matter [1].

## Figures and Tables

**Figure 2 materials-15-03901-f002:**
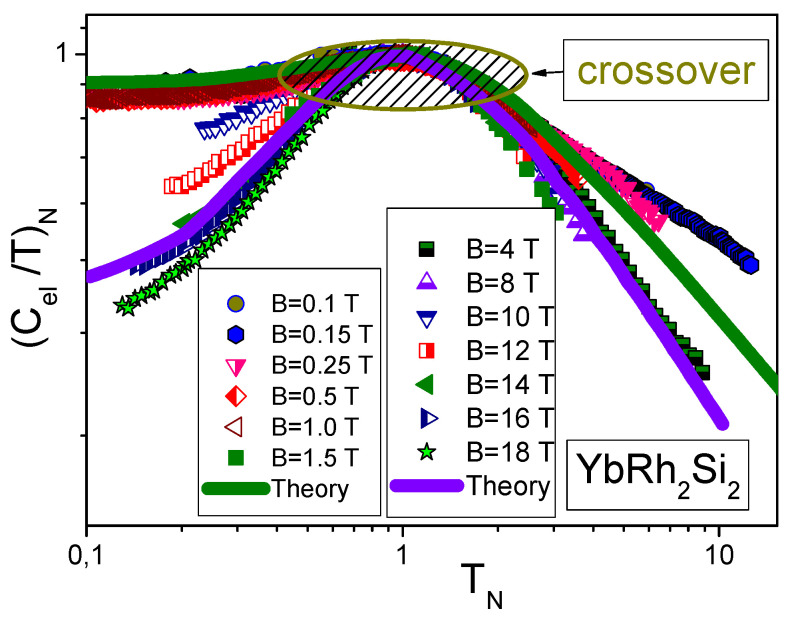
The normalized specific heat ${\left(C_{\mathrm{el}}/T\right)}_{N}=M_{N}^{*}$ of ${\mathrm{YbRh}}_{2}{\mathrm{Si}}_{2}$ as a function of normalized temperature $T_{N}$ under the application of magnetic field *B* shown in the left hand legend (low *B*) and the right hand legend (high *B*) The experimental data is extracted from the measurement of C/T measurements on the archetypical HF metal ${\mathrm{YbRh}}_{2}{\mathrm{Si}}_{2}$ [52,53]. The low-field calculations of $M_{N}^{*}$ are depicted by the solid green curve. The solid blue curve representing high-field calculations (B∼18) is performed for the fully polarized quasiparticle band [29]. Starting from relatively high magnetic fields B≥4 T, the specific heat demonstrates the same behavior as $\left(C_{\mathrm{mag}}/T\right)=M_{N}^{*}$ of ${\mathrm{Lu}}_{3}{\mathrm{Cu}}_{2}{\mathrm{Sb}}_{3}O_{14}$ shown in Figure 1.

**Figure 3 materials-15-03901-f003:**
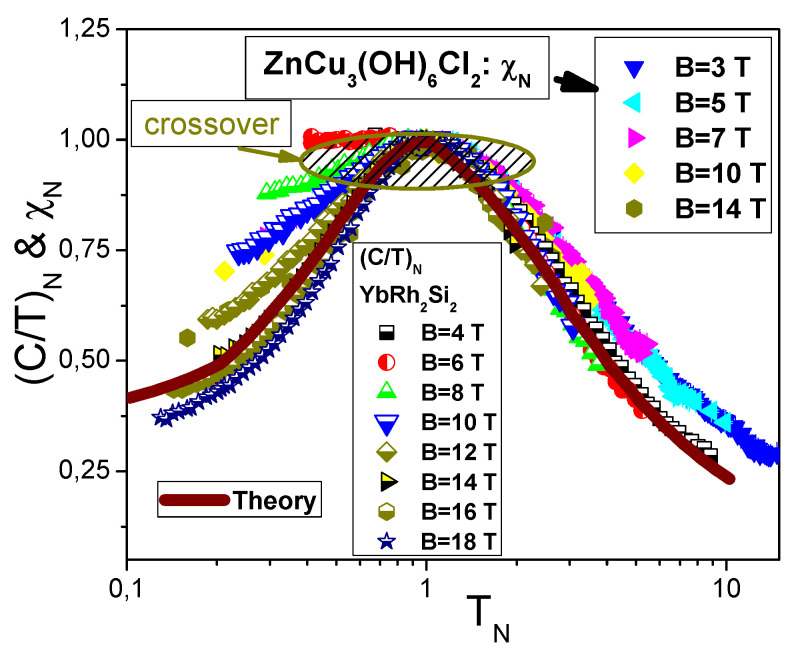
Normalized magnetic susceptibility $\chi_{N}=\chi /\chi_{\max}=M_{N}^{*}$ as a function of the normalized temperature $T_{N}\propto T/B$. The values of magnetic field *B* are displayed in the legend. The data are extracted from the measurements of the magnetic susceptibility χ(T,B) on ${\mathrm{ZnCu}}_{3}{\left(\mathrm{OH}\right)}_{6}{\mathrm{Cl}}_{2}$ [16]. The normalized data of ${\left(C_{\mathrm{el}}/T\right)}_{N}=M_{N}^{*}$ are obtained from the specific heat measurements $C_{\mathrm{el}}/T$ of ${\mathrm{YbRh}}_{2}{\mathrm{Si}}_{2}$ in the presence of magnetic field *B* (the legend) [52]. The solid curve represents theoretical calculations at B≃18 T at the fully polarized quasiparticle band. It demonstrates universal scaling of $M_{N}^{*}$, and coincides with the universal scaling of the quantum spin liquid depicted in Figure 1. The crossover from the LFL behavior to the NFL one is displayed by the arrow.

**Figure 4 materials-15-03901-f004:**
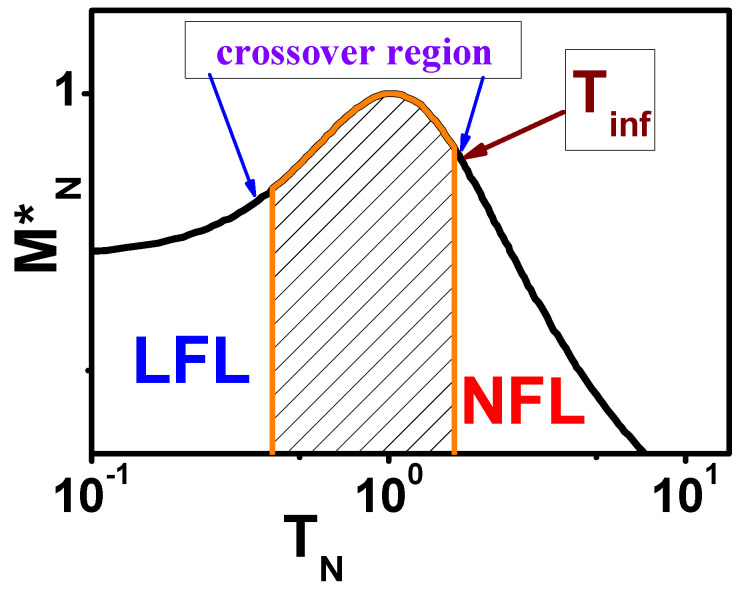
Schematic plot of the normalized effective mass $M_{N}^{*}=M^{*}/M_{M}^{*}$ versus the normalized temperature $T_{N}=T/T_{M}$. The crossover region, at which $M_{N}^{*}$ reaches its maximum $M_{N}^{*}=1$ at $T_{N}=1$, is shown by the arrows. The inflection point $T_{inf}$ at which the system enters the crossover region is displayed by the arrow. The LFL and NFL regions are labeled.

**Figure 5 materials-15-03901-f005:**
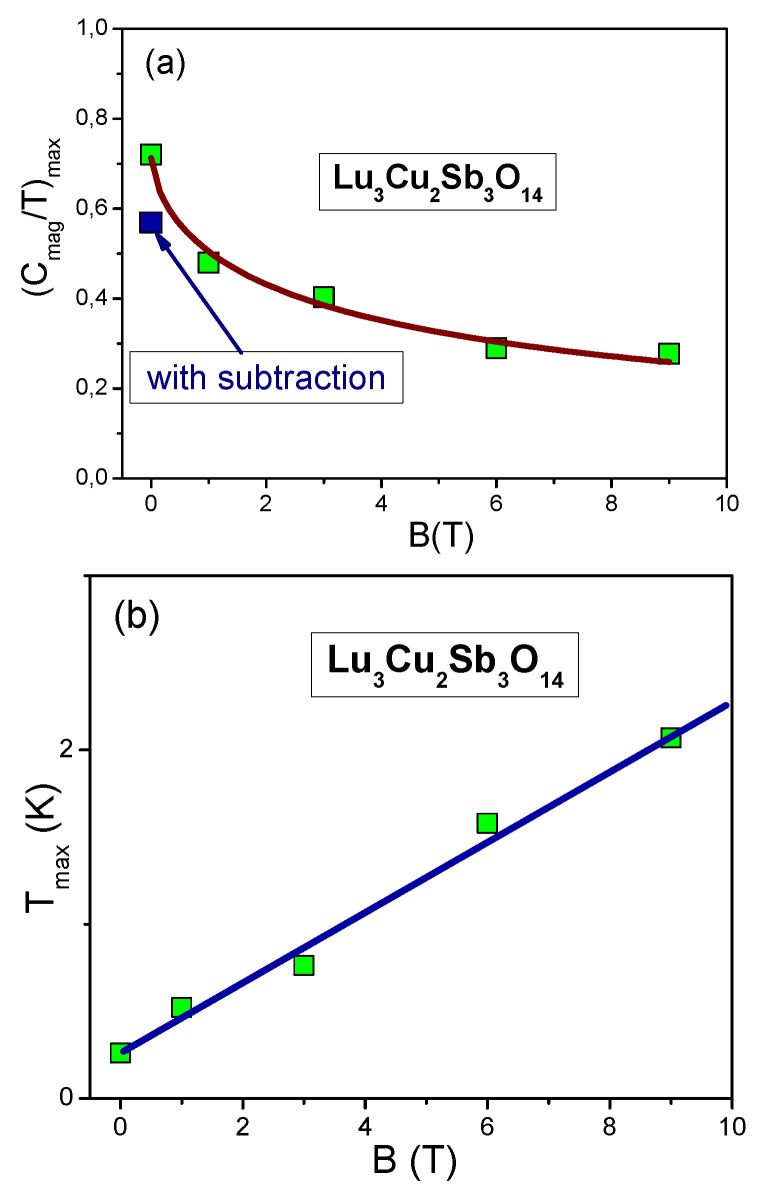
The properties of the specific heat. (**a**) The maximum values of ${\left(C_{\mathrm{mag}}/T\right)}_{\max}$ of the specific heat $C_{\mathrm{mag}}/T$ versus magnetic field *B* are shown by the solid squares, see the inset, Figure 1. The solid curve is approximated by $M_{\max}^{*}\left(B\right)\propto B^{-2/3}$ in accordance with Equation (Equation 5). The arrow depicts the position of ${\left(C_{\mathrm{mag}}/T\right)}_{\max}$ at B=0 when the impurity Schottky contribution is subtracted [23]. (**b**) The temperature $T_{\max}\left(B\right)$, at which the maxima of $\left(C_{\mathrm{mag}}/T\right)$ are located, see Figure 1. The solid straight line traces the function $T_{\max}\propto B$, see Equation (Equation 7).

**Figure 6 materials-15-03901-f006:**
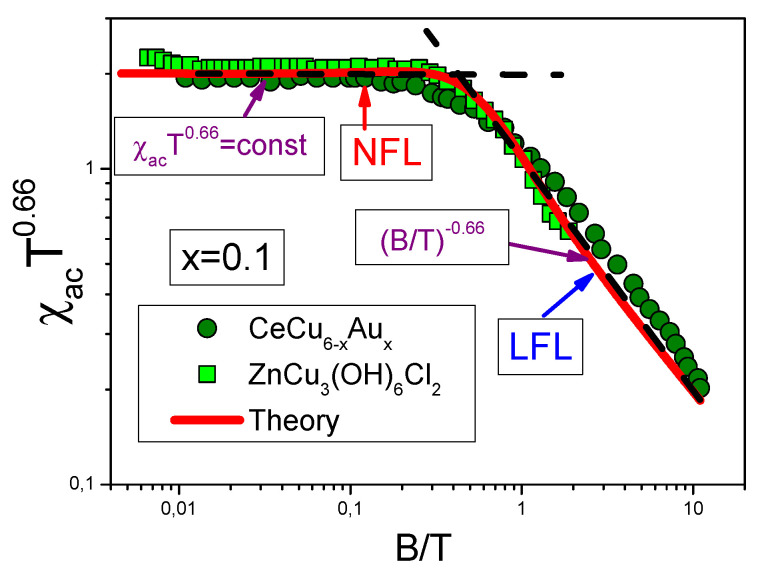
The universal scaling behavior of different strongly correlated Fermi systems versus B/T. The universal behavior of the HF metal ${\mathrm{CeCu}}_{6-x}{\mathrm{Au}}_{x}$ is extracted from data [63], and that of ${\mathrm{ZnCu}}_{3}{\left(\mathrm{OH}\right)}_{6}{\mathrm{Cl}}_{2}$ is derived from data [16]. At B/T≪1 the systems exhibit the NFL behavior, that is $T^{2/3}\chi \propto const$. At B/T≫1 the systems demonstrate the LFL behavior, with χ being a decreasing function of B/T.

**Figure 7 materials-15-03901-f007:**
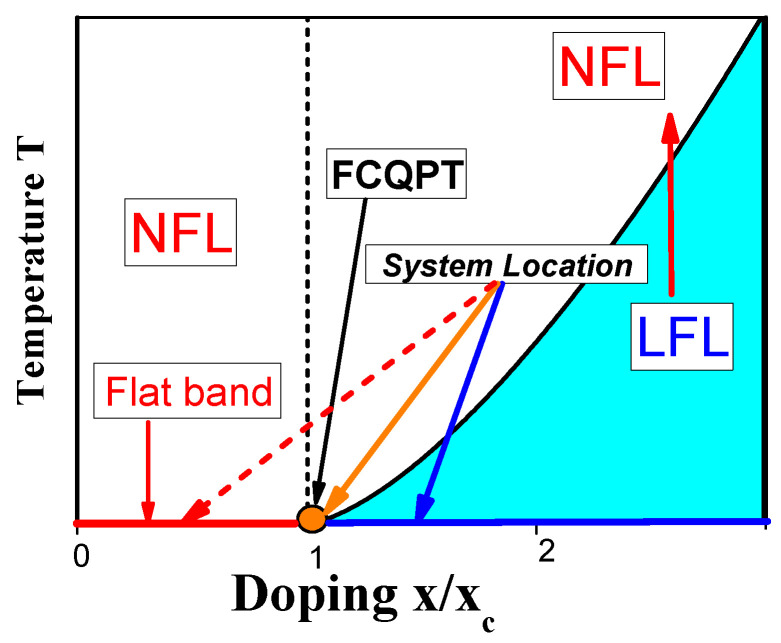
Schematic temperature *T*—doping $x/x_{c}$ phase diagrams of HF metals. The number density *x* is taken as the control parameter and depicted as $x/x_{c}$. At $x/x_{c}<1$ the dashed arrow shows the the ordered phase of the topological FCQPT, when the system possesses flat bands. At any finite temperature T>0 and at $x/x_{c}<1$, the system exhibits the NFL. The shadowed area corresponds to the case $x/x_{c}>1$ and sufficiently low temperatures, where the system is in the LFL phase.

**Figure 8 materials-15-03901-f008:**
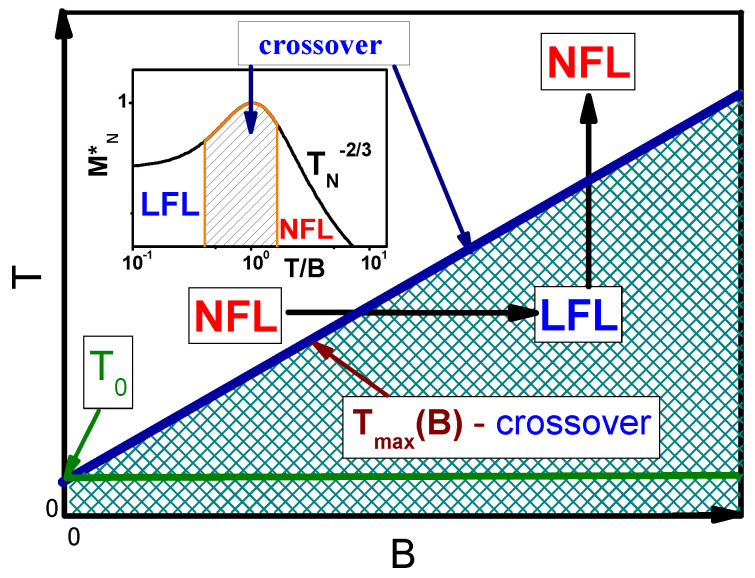
Schematic T−B phase diagram of ${\mathrm{Lu}}_{3}{\mathrm{Cu}}_{2}{\mathrm{Sb}}_{3}O_{14}$ with magnetic field as the control parameter. The vertical and horizontal arrows depict LFL-NFL transitions at fixed *B* and *T*, respectively. The line separating LFL-NFL regions is shown by the arrow, representing the crossover region at $T_{\max}\left(B\right)$ and displaying the function $T_{\max}\left(B\right)$. $T_{0}$ is the temperature at which the LFL behavior occurs. The inset demonstrates a chart of the normalized effective mass $M_{N}^{*}$ as a function of $T_{N}\propto T/B$. Transition region is characterized by the maximum value $M_{\max}^{*}$ of $M^{*}$ at $T_{N}=T/T_{\max}=1$.

**Figure 9 materials-15-03901-f009:**
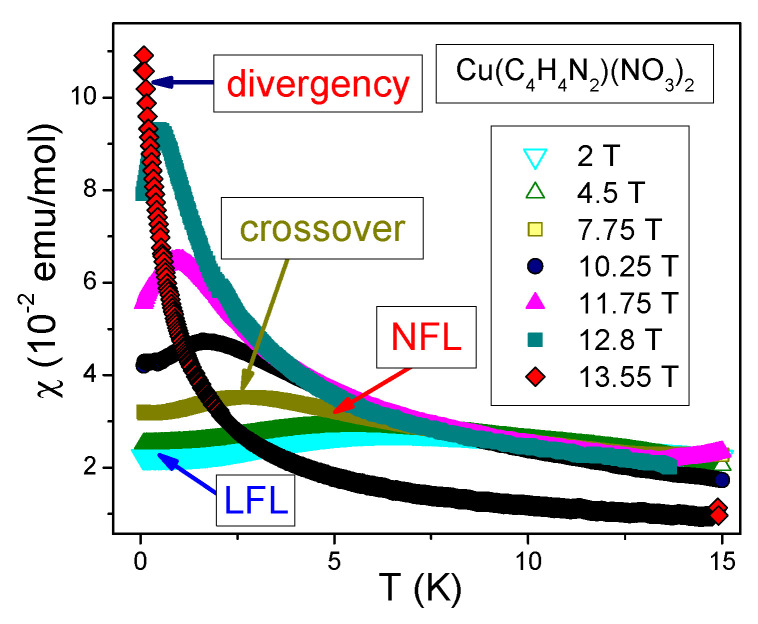
Temperature dependence of χ(T) for CuPzN. The experimental data are from [25]. The magnetic fields are shown in the legend. The three regions LFL, NFL and crossover are shown. The divergent behavior of χ(T) at $B=B_{s}$ is indicated by the black arrow.

**Figure 10 materials-15-03901-f010:**
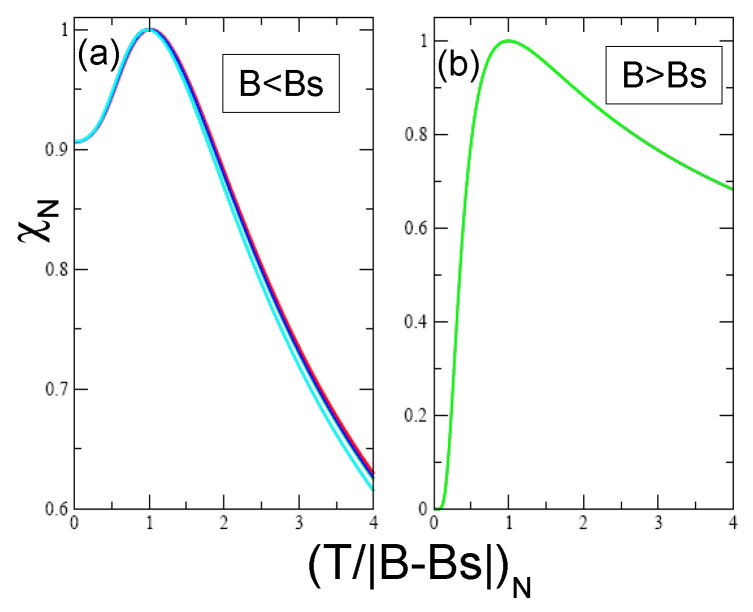
The normalized magnetic susceptibility $\chi_{N}$ vs. the normalized $(T/|B-B_{s}{\left|\right)}_{N}$, calculated from Equations (Equation 12) and (Equation 13) for $B<B_{s}$ (panel (**a**)) and $B>B_{s}$ (panel (**b**)). At $B>B_{s}$ the complete spin polarization occurs and χ vanish at ${\left(T/B\right)}_{N}\to 0$. Two different curves on panel (**a**) show an excellent scaling of $\chi_{N}{\left(T/B\right)}_{N}$. As seen from panel (**a**), the LFL behavior holds for the weakly interacting quasi-1D quantum spin liquid [22].

**Figure 11 materials-15-03901-f011:**
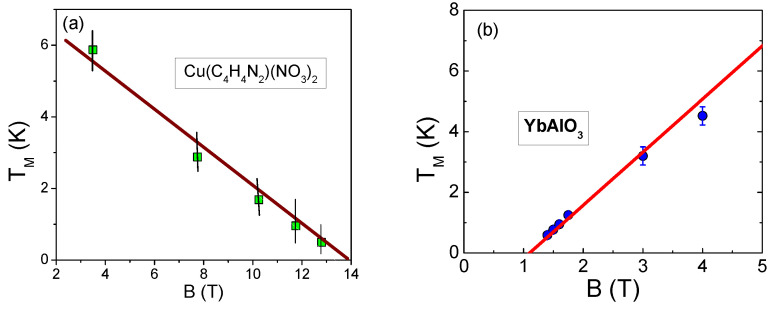
The magnetic field dependence of peak temperature $T_{M}\left(\chi \right)$. Panel (**a**) shows the $B-T_{M}$ dependence of CuPzN, extracted from data [25]. Panel (**b**) demonstrate the $B-T_{M}$ dependence for ${\mathrm{YbAlO}}_{3}$, extracted from experimental data [24]. The calculated straight lines $T_{M}=a\left|B_{s}-B\right|$ given by Equation (Equation 17). The excellent coincidence is seen, showing that Equation (Equation 17) holds for both CuPzN and ${\mathrm{YbAlO}}_{3}$.

**Figure 12 materials-15-03901-f012:**
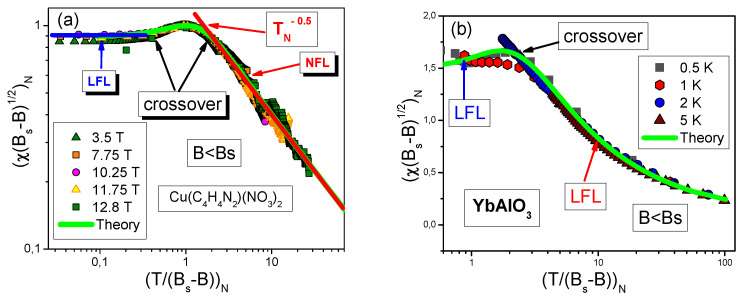
The normalized magnetic susceptibility $\chi_{N}$ extracted from measurements at $B<B_{s}$. Panel (**a**): $\chi_{N}$ extracted from measurements on CuPzN [25]. Panel (**b**): $\chi_{N}$ extracted from measurements on ${\mathrm{YbAlO}}_{3}$ [24]. Our theoretical curve, plotted based on Equations (Equation 12) and (Equation 14), is represented by the solid line (shown in Figure 10a) tracing the scaling. It is seen that the dependence $\chi_{N}$ for ${\mathrm{YbAlO}}_{3}$ has three regions: LFL, crossover and NFL.

**Figure 13 materials-15-03901-f013:**
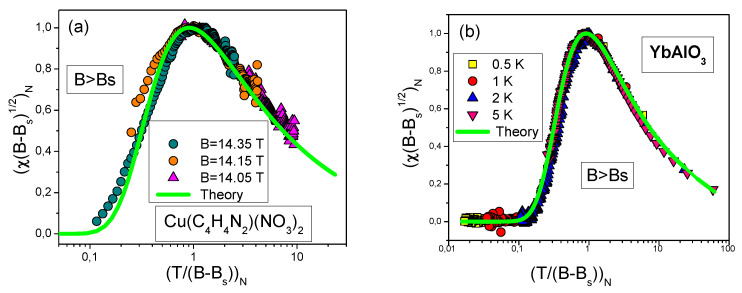
The normalized magnetic susceptibility $\chi_{N}$ extracted from measurements in magnetic fields $B>B_{s}$ shown in the legend. Panel (**a**): $\chi_{N}$ extracted from measurements on CuPzN [25]. Panel (**b**): $\chi_{N}$ extracted from measurements on ${\mathrm{YbAlO}}_{3}$ [24]. Our theoretical curves, plotted on the base of Equations (Equation 12) and (Equation 14), is taken from Figure 10b, and is reported by the solid line tracing the scaling behavior.

**Figure 14 materials-15-03901-f014:**
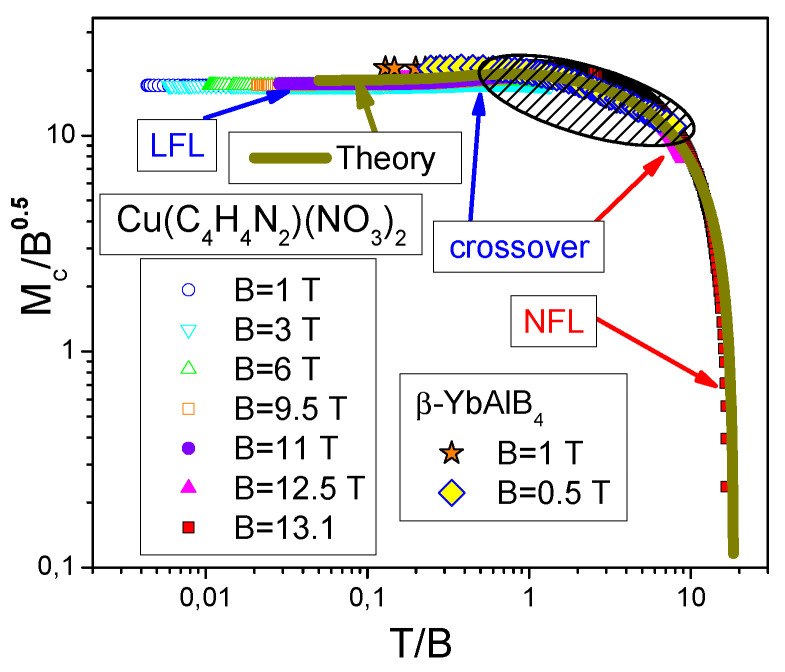
The scaling dependence of magnetization normalized to maximal values ${\left(M_{c}/B^{1/2}\right)}_{N}$ ($M_{c}=M_{s}-M$, $B=B_{s}-B$) on ${\left(T/B\right)}_{N}$ for CuPzN and β$-{\mathrm{YbAlB}}_{4}$. The experimental data are taken from [25,31]. The magnetic fields *B* (in T) are shown in the legends. The typical LFL and NFL behavior are shown by both the arrows and the straight lines, the crossover is displayed by the arrow. The theory is represented by the solid green curve [22,43].

**Figure 15 materials-15-03901-f015:**
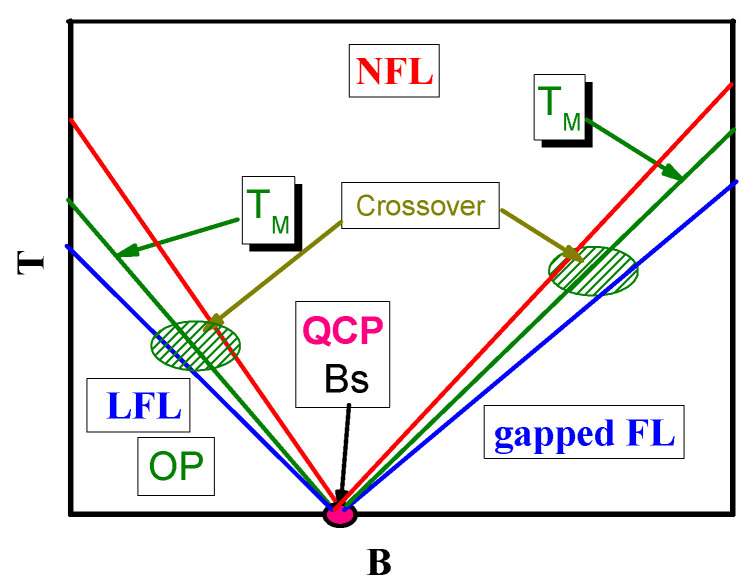
Schematic magnetic field—temperature phase diagram of 1DQSL. Straight lines on both sides of $B_{s}$, which is a FCQPT point, indicate, respectively, the lines of LFL boundary (the lowest temperature), the temperatures of maxima (middle line, marked “$T_{M}$” taken from Figure 11a) and the end of crossover region: The highest temperature at which the system enters the NFL regime. The left sector labeled as “LFL” and “OP” (ordered phase) displays the LFL behavior and possible ordered phase of spin liquid. The right sector labeled as “gapped FL” denotes the gapped field-induced ferromagnetic spin liquid.

**Figure 16 materials-15-03901-f016:**
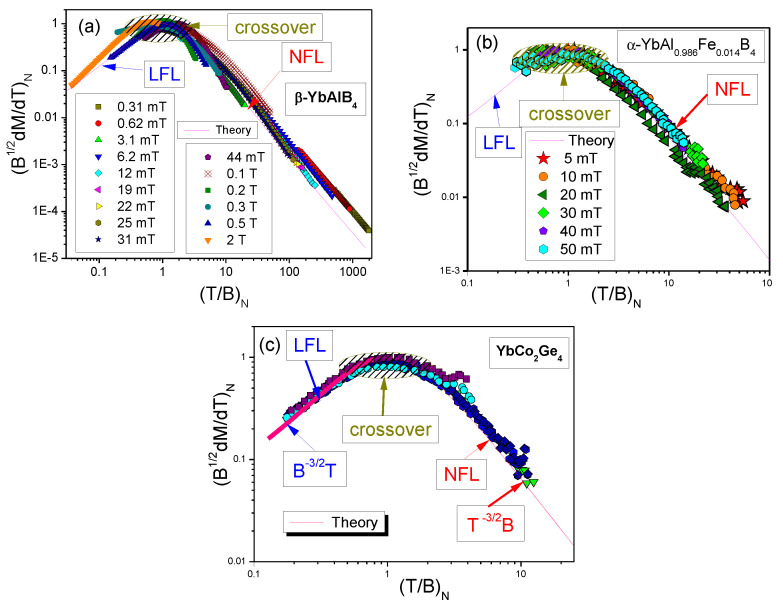
Scaling of dimensionless normalized magnetization ${\left(B^{1/2}dM\left(T,B\right)/dT\right)}_{N}$ as a function of the dimensionless normalized ${\left(T/B\right)}_{N}$ at magnetic field values *B* given in the legends (**a**,**b**). Regions of LFL behavior, crossover, and NFL behavior are indicated by arrows. The theory is represented by the solid curve [14,43]. Data are extracted from [32,33,34]. Panel (**a**): Scaling of β$-{\mathrm{YbAlB}}_{4}$ [43]. Panel (**b**): Scaling of α$-{\mathrm{YbAl}}_{1-x}{\mathrm{Fe}}_{x}B_{4}$. Panel (**c**): Scaling of ${\mathrm{YbCo}}_{2}{\mathrm{Ge}}_{4}$, measured at different field values B=0.05,0.1,0.2,0.3,0.5 T [33].

**Figure 17 materials-15-03901-f017:**
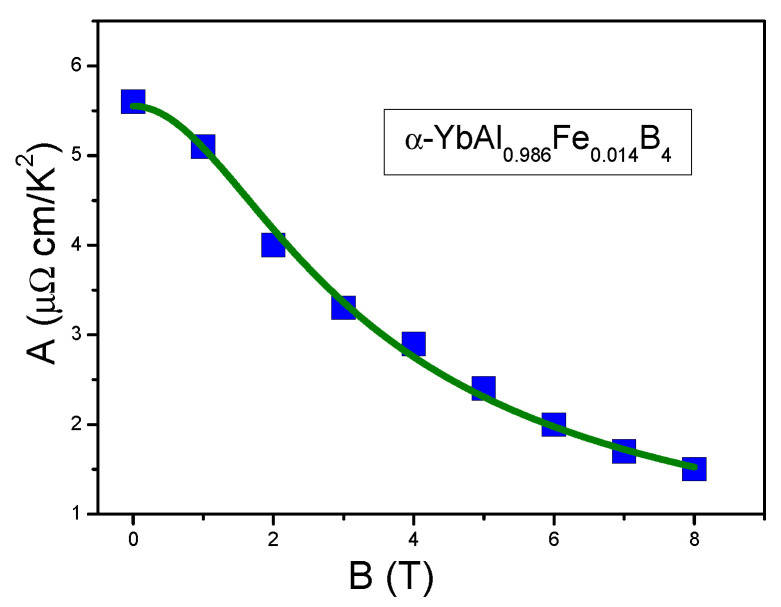
Experimental data for the coefficient A(B), plotted as a function of magnetic field *B* (solid squares). Measured values of A(B) are extracted from the experimental data [34]. The solid curve is given by Equation (Equation 28).

**Figure 18 materials-15-03901-f018:**
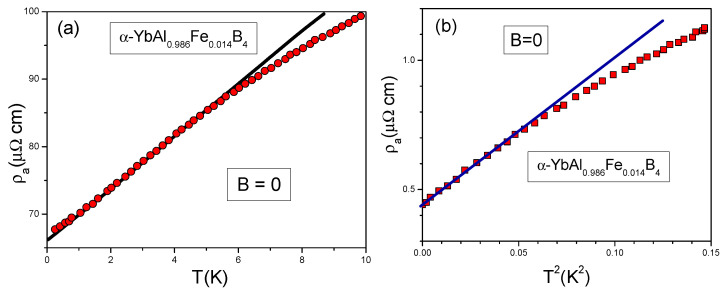
Experimental data for the resistivity $\rho_{a}\left(T\right)$: in panel (**a**) resistivity is plotted as a function of *T*, solid circles and in panel (**b**) resistivity is plotted as a function of $T^{2}$, solid squares. The data are taken from [34]. The solid lines are fits to the experimental data.

**Figure 19 materials-15-03901-f019:**
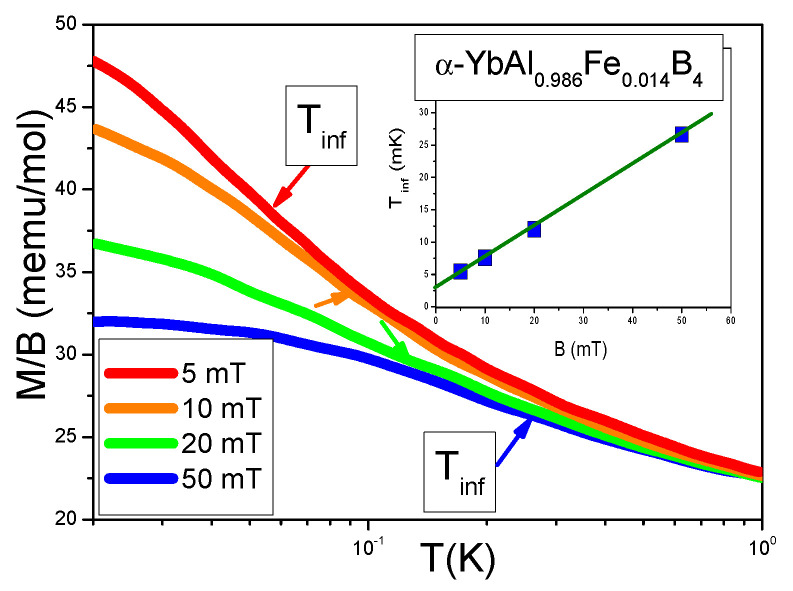
Magnetization M(T) versus *T* for *B* values shown in the legend. The data are taken from [34]. M(T) is displayed versus temperature on a logarithmic scale. The approximate location of the inflection point at temperature $T_{inf}\left(B\right)$ (versus magnetic field *B*) is indicated by the arrow. At $T=T_{inf}\left(B\right)$ the HF metal α$-{\mathrm{YbAl}}_{1-x}{\mathrm{Fe}}_{x}B_{4}$ enters the crossover region, separating the NFL behavior from the LFL one, see Figure 7. The inset displays $T_{inf}$ versus magnetic field *B*.

## Data Availability

Not applicable.

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
