# Peer review of "Strongly Correlated Quantum Spin Liquids versus Heavy Fermion Metals: A Review"

_materials, 2022, doi:10.3390/ma15113901_

Round 1

Reviewer 1 Report

Referee Report on 
"Strongly correlated quantum spin liquids versus heavy fermion metals: a review"
by V. R. Shaginyan, A. Z. Msezane, G. S. Japaridze, S. A. Artamonov, and Y. S. Leevik

    In this review article the authors discuss recent experimental results related to physics of Heavy Fermion (HF) compounds. The experimental data are interpreted in within the framework of fermion-condensation quantum phase transition theory. Additional arguments are presented supporting the conclusion from the previous papers of the authors the HF materials form a new state of matter. Another realization of that state is provided by quantum spin liquids. The paper discusses the dependence of thermodynamic and transport properties of HF compounds on temperature and on the external magnetic field. The transitions between LFL and NFL states are discussed as well. The paper covers an interesting topic and I recommend its publication.

Author Response

We thank the referee for the report.

Reviewer 2 Report

This is review article and one expects all notions to be defined. This is not the case, for example fermion condensation, flat band, spin liquid .....

This is review and one expects that not only authors papers are cited. For example papers for flat band.

Author Response

We thank the referee for the report, which is to improve our review. 

We have taken into account all the referee’s criticism: on p. 2 we have added Eqs. (1) and (2)  clarifying flat bands and the Fermion condensation and explained the notion of  spin liquid. 

We have also added Refs [8,9] on flat bands.  All the corrections in revised version are highlighted by green.   

Reviewer 3 Report

This paper is devoted to a review on strongly correlated quantum spin liquid vs heavy fermion metals. The Authors present a series of results, both theoretical and experimental, on some specific materials trying to find a unified picture, so explaining experimental results. Although this approach would be strategically worth to apply, it is here implemented in a very confused manner. Indeed,

1) the title of the paper is misleading since it refers to a large class of materials, but the paper is devoted to some specific compounds;

2) the aim of the paper is not clearly stated since, in my opinion, the flow of the manuscript is not completely understandable. The experimental results are presented together with some theoretical ones, and it is hard to find a unified picture. Moreover, I argue that the Authors would like to discuss, separately, quasi-one dimensional quantum spin liquids an then heavy fermion metals, but this approach is not correct if they claim that the paper compares the two systems.

Therefore, the paper cannot be accepted in the present form; before publication it must be largely reorganized, and its aim clearly stated.

Author Response

We thank the referee for the report.

The referee writes:

“The Authors present a series of results, both theoretical and experimental, on some specific materials trying to find a unified picture, so explaining experimental results. Although this approach would be strategically worth to apply, it is here implemented in a very confused manner. Indeed,

1) the title of the paper is misleading since it refers to a large class of materials, but the paper is devoted to some specific compounds”

We disagree with this statement. The title of our review is very clearly written: “Strongly correlated quantum spin liquids versus heavy fermion metals: a review”. We show that quantum spin liquids (QSL) and heavy fermion metals have the same behavior with one exception: QSL cannot support charger current.  In our short review, we consider new experimental facts recently collected on the corresponding compounds. While the other numerous experimental facts are considered in our recent reviews, see e.g. [13,14]. We show that these compounds can be described within the framework of the fermion condensation theory by explaining the thermodynamic properties of compounds.

The referee continues:

2) the aim of the paper is not clearly stated since, in my opinion, the flow of the manuscript is not completely understandable. The experimental results are presented together with some theoretical ones, and it is hard to find a unified picture. Moreover, I argue that the Authors would like to discuss, separately, quasi-one dimensional quantum spin liquids an then heavy fermion metals, but this approach is not correct if they claim that the paper compares the two systems.

We disagree with this statement. First of all, we intently present the theoretical consideration along with analyzing the experimental facts. Obviously, a theory has to be supported by the corresponding experimental facts, for it is the basement of physics. Then, we clearly demonstrated that both quasi-1D QSL and HF metals (like YbNi4P2) exhibit the same behavior as 3D HF metals, see Figs. 11-13 that demonstrate fine agreement between experimental facts and our theory. Thus, such a consideration enforces our consideration. As a result, there is no need to reorganize our review, while its aim is clearly stated.

Reviewer 4 Report

In this Review papers, the authors try to do connections and comparisons between the Quantum spin liquids and Heavy fermion systems in an harmoniously theoretical framework. The author also points out their similar scaling behavior near FC quantum phase transition under magnetic field. This paper is quite useful and provide new viewpoint for the frustrated magnets and deserved to be published in Materials. I appreciated the authors can reply the following comments before publication.

  • On page 2, “HF compounds like Lu3Cu2Sb3O14”, this compound is not confirmed to by heavy Fermion system but spin liquid, should modify this expression.
  • Lu3Cu2Sb3O14 is called “ultra spin liquid” not spin liquid or quantum spin liquid, can the authors provide the explanations.
  • The title for Both “V and ” is “ phase diagram”, I suggest to make a differences

Author Response

We thank the referee for the report which is to improve our review.

We have taken into account all the referee’s criticism: we have eliminated the term “ultra spin liquid” and changed “HF compounds like Lu3Cu2Sb3O14” to “the frustrated insulator”.

By the way, the term “ultra spin liquid” is taken from Ref. [23].  We have changed the title of both Sections V and VII. Now these are named “Phase diagram of one dimensional quantum spin liquids“and “Schematic temperature - doping phase diagram”, correspondingly.